# Cache-Augmented Inbatch Importance Resampling for Training Recommender Retriever

**Jin Chen**[1], **Defu Lian**[2]*, **Yucheng Li**[2], **Baoyun Wang**[3], **Kai Zheng**[1], **Enhong Chen**[2]

[1]University of Electronic Science and Technology of China
[2]University of Science and Technology of China, [3]Hisense
chenjin@std.uestc.edu.cn, liandefu@ustc.edu.cn, liycustc@mail.ustc.edu.cn,
wangbaoyun@hisense.com, zhengkai@uestc.edu.cn, cheneh@ustc.edu.cn

## Abstract

Recommender retrievers aim to rapidly retrieve a fraction of items from the entire item corpus when a user query requests, with the representative two-tower model trained with the log softmax loss. For efficiently training recommender retrievers on modern hardwares, inbatch sampling, where the items in the mini-batch are shared as negatives to estimate the softmax function, has attained growing interest. However, existing inbatch sampling based strategies just correct the sampling bias of inbatch items with item frequency, being unable to distinguish the user queries within the mini-batch and still incurring significant bias from the softmax. In this paper, we propose a Cache-Augmented Inbatch Importance Resampling ($\mathcal{X}$IR) for training recommender retrievers, which not only offers different negatives to user queries with inbatch items, but also adaptively achieves a more accurate estimation of the softmax distribution. Specifically, $\mathcal{X}$IR resamples items from the given mini-batch training pairs based on certain probabilities, where a cache with more frequently sampled items is adopted to augment the candidate item set, with the purpose of reusing the historical informative samples. $\mathcal{X}$IR enables to sample query-dependent negatives based on inbatch items and to capture dynamic changes of model training, which leads to a better approximation of the softmax and further contributes to better convergence. Finally, we conduct experiments to validate the superior performance of the proposed $\mathcal{X}$IR compared with competitive approaches.

## 1 Introduction

Recommender systems (RS) [9, 10, 23, 24] have been prevalently developmented to address the issue of information overload, with the purpose of mining potential preferred items for users, bringing great commercial revenues to enterprises. Recommender retrievers, which retrieve a subset of highly relevant items from numerous items, are a key component of the commonly-used two-stage architecture [9, 13, 34]. The two-tower models, where the queries and items are represented with separate models, are representative of retrieval models. Generally, given the similarity scores of user and item embeddings, the models attempt to minimize the loss function [10, 17], i.e., the log-softmax loss, guiding the interacted items having higher similarity scores. However, it is extremely inefficient to calculate the partition function in the log-softmax loss over the item corpus. Negative Sampling, which samples a number of items to estimate the partition function or the gradient, has attained growing interests in recommender retrievers [16, 25, 31, 34, 35].

A straightforward method is to take the static uniform distribution as the proposal distribution to sample items from the entire item set. Nevertheless, restricted by the computational resource, a more

---

*Corresponding author

36th Conference on Neural Information Processing Systems (NeurIPS 2022).

practical strategy is *inbatch sampling* [15, 34], where the other training samples in the mini-batch are shared as negative samples. Considering the complicated features in online systems, it is time-consuming to get the embeddings of users and items via the tower model. Encoding the sampled unseen items outside the batch occupies additional huge computational cost. Thus, the inbatch sampling is more efficient and profitable for online systems under the circumstances of significantly great number of users and items with numerous features. There exists a sampling bias of training data in inbatch sampling, since the interactions are received from user logs and the item exposure follows the long-tailed distribution. Accordingly, several works correct the sampling bias using the popularity of items [35, 37] and have been successfully exploited into the online systems.

However, there still exists a huge bias of the approximated gradient from the full softmax with the inbatch sampling. The sampling strategies eventually fit in the sampled-softmax, which approximates the exact gradient of softmax loss depending on the importance sampling. The bias can be eliminated by forcing the proposal distribution, i.e., sampling distribution, to the softmax distribution [2]. Previous works [34, 35] indeed correct the sampling distribution for inbatch sampling as the frequency-based distribution (or popularity-based distribution), which remains a query-independent proposal. The ideal softmax distribution is dependent on different queries and changes dynamically during the training process. Existing studies [12, 22, 33, 36, 38] in negative sampling for recommender systems have proved that adaptive samplers lead to better convergence of the recommendation quality. Thus, there is an urgent need to develop an adaptive selection strategy for inbatch sampling to reduce the bias towards the ideal softmax distribution.

To this end, we propose a Cache-Augmented Inbatch Importance Resampling, shorted as $\mathcal{X}$IR, for training recommender retrievers. Concretely, we resample the batch items according to the modified softmax weight over the mini-batch, which offers different negatives to user queries, instead of sharing a same set of batch items. This importance resampling based selection not only distinguish users but also provides a more accurate estimation of the full softmax. Furthermore, inspired by the recent work [34], which introduces additional uniformly sampled negatives to tackle the sampling bias, a cache mechanism is utilized to further eliminate the approximation bias caused by the sampling bias of the inbatch sampling. Rather than the uniformly sampled items, we cache the more frequently sampled items into a fixed-size set, usually having the same size as the mini-batch. The frequently sampled items tend to have higher similarity scores on average, which can be regarded as *hard* negative samples for training models. Therefore, the cache plays a role of item augmentation since more informative negatives are introduced to accelerate the model convergence. The proposed $\mathcal{X}$IR is then evaluated on five public real-world datasets, demonstrating the superior performance over competitive baseline approaches in terms of effectiveness.

## 2 Preliminaries

### 2.1 Recommender Retrievers

Given the user query, recommender retrievers aim to rapidly response with highly relevant items from the full item corpus. Under the paradigm of the two-tower models [5, 21, 30], the user queries and items are represented by $\{\boldsymbol{q}_i \in \mathbb{R}^{d_u}\}_{i=1}^{M}$ and $\{\boldsymbol{e}_j \in \mathbb{R}^{d_i}\}_{j=1}^{N}$, where $M$ and $N$ denote the number of queries and items, respectively. Here, we assume that the number of queries is finite for simplification. The user tower and item tower then map the feature vectors to a $k$-dimensional embedding space with separate functions $\phi_Q : \mathbb{R}^{d_u} \to \mathbb{R}^k$ and $\phi_I : \mathbb{R}^{d_i} \to \mathbb{R}^k$, respectively. A similarity scorer $s$, such as the widely used inner product function $s(u, i) = \langle \phi_Q(\boldsymbol{q}_u), \phi_I(\boldsymbol{e}_i) \rangle$, outputs the relevance of the query $u$ and candidate item $i$. In this paper, we investigate the log-softmax loss for training the two-tower models, where the preference probability of the item $i$ with respect to the query $u$ is calculated with $P(i|u) = \frac{\exp(s(u,i))}{\sum_{j \in \mathcal{I}} \exp(s(u,j))}$, where $\mathcal{I}$ denotes the candidate item corpus. By maximizing the likelihood, the objective function follows as:

$$\mathcal{L}_{\text{softmax}}(\mathcal{D}, \Theta) = -\frac{1}{|\mathcal{D}|} \sum_{(u,i) \in \mathcal{D}} \log P(i|u) = -\frac{1}{|\mathcal{D}|} \sum_{(u,i) \in \mathcal{D}} \log \frac{\exp(s(u,i))}{\sum_{j \in \mathcal{I}} \exp(s(u,j))} \tag{1}$$

where $\Theta$ denotes the model parameters in $\phi_Q$ and $\phi_I$. $\mathcal{D}$ denotes the training data, each tuple of which records an interaction between the user query $u$ and the item $i$. The two-tower model is such trained and the interacted items, i.e., positive pairs, are encouraged to get higher scores than

uninteracted items. The maximum inner product search (MIPs) then can be fruitfully implemented, such as SPANN [8], ScaNN [14] and FAISS [20], to search for the topk items given a user query, meeting the requirements of rapid online response.

When the number of items $N$ becomes extremely large, it would incur a substantial computational cost of the partition function, i.e., the denominator in Eq (1). Intuitively, the partition function can be estimated by sampling a subset from candidate items. The sampled softmax loss then behaves as the objective function,

$$\mathcal{L}_{\text{sampled\_softmax}}(\mathcal{D}, \Theta) = -\frac{1}{|\mathcal{D}|} \sum_{(u,i) \in \mathcal{D}} \log \frac{\exp\left(s'\left(u,i\right)\right)}{\sum_{j \in \mathcal{S}} \exp\left(s'\left(u,j\right)\right)} \tag{2}$$

where $\mathcal{S}$ represents the subset of the sampled items. $s'\left(u,i\right) = s\left(u,i\right) - \log p(i|u)$ denotes the corrected logit with the sampling probability. Obviously, the computational cost is considerably reduced, which only depends on the size of the sampled subset rather than the total item corpus.

## 2.2 Negative Sampling in RS

To guarantee the unbiased gradient estimation of the sampled softmax, the softmax distribution is the ideal proposal distribution [2, 18], which is prohibitively expensive given the huge number of candidates and features. To meet the requirements of quick response, the query-independent sampling distribution, such as the uniform distribution, is applied to sample items. These distributions remain constant during the dynamic model training and they will select easy samples in later training epochs, which contribute a limited gradient to model convergence. The adaptive samplers are further investigated to sample more informative items depending on the model and can be grouped into two categories. One leverages techniques of adversarial frameworks [11, 26, 32], where negative samples are generated through the complicated networks. The other category attempts to assign a higher probability to items with larger similarity scores [19, 29, 36] or to reduce the bias of the sampling distribution from the softmax distribution [4, 6, 7, 22]. One promising approach is the two-pass sampler [1, 36], which samples a fixed-sized pool of items from the simple static distribution and then selects the items with the highest similarity scores in the pool to calculate the loss. Due to its intractable sampling distribution, it is seldom cooperated within the sampled softmax loss and would perform worse due to its huge approximation bias from the softmax distribution.

However, in the online service of recommender systems with a huge magnitude of items and features, the query and item embeddings are constantly calculated through the large-scale deep towers. The inbatch sampling becomes popular since it avoids feeding additional negative items to the item tower [34], saving much computational cost. Moreover, the inbatch sampling does not improve the computational costs from the complexity perspective [28] but lowers the time cost on modern hardwares. To reduce the bias caused by the item exposure in online systems, several works correct the bias with item frequency which fits the sampled softmax loss with the item frequency as the proposal [10, 34, 35, 37].

# 3 Training Recommender Retrievers with Cache-Augmented Inbatch Importance Resampling

The inbatch sampling has attracted more attention due to its high efficiency for deployment on modern hardwares in online systems. Existing works attempt to correct the sampling bias within the mini-batch by the item frequency, which finally acts as the popularity-based proposal distribution in the sampled softmax loss. However, considering the ideal softmax distribution, whose sampling probability is associated with the given query, there still exists a bias of gradient estimation from the softmax by just correcting the selection bias of inbatch sampling. To achieve a better approximation of the softmax, we propose the inbatch importance resampling (BIR for short) and the cache-augmented importance resampling ($\mathcal{X}$IR), which not only delivers different negative items to different queries based on the inbatch items but also yields an adaptive softmax estimation during model training.

## 3.1 Reducing Bias of Inbatch Sampling based on Importance Resampling

Items within the mini-batch actually follow the popularity-based distribution since the observed items are exposed to users under the long-tailed distribution in today's recommender systems. Thus,

a query-independent sampler, i.e., frequency-based sampler, fits the sampled softmax with the inbatch sampling. However, this query-independent sampler corrects the selection bias of the inbatch sampling but does not improve the estimation performance.

The importance resampling [3] is adopted here to further correct the bias from the ideal softmax distribution. Given the mini-batch $B$, each item is resampled for the query $u$ based on the weight

$$w(i|u) = \frac{\exp\left(s(u,i) - \log pop(i)\right)}{\sum_{j \in B} \exp\left(s(u,j) - \log pop(j)\right)},$$

where the probability $pop(i)$ here denotes the frequency of the items summarized from the training data. The resampled item set with respect to the query $u$ is denoted as $\mathcal{R}_u$.

By applying the resampling strategy, items with larger similarity scores have a higher chance of being sampled and may appear multiple times. Moreover, the sampled items for different user queries are different while the queries share the same item set in inbatch sampling. The following theorem provides the evidence that the sampled items based on the inbatch importance resampling can accurately approximate the softmax distribution.

**Lemma 3.1.** *Assume items follow a distribution $P = \{p_1, p_2, ..., p_N\}$, the item set $\mathcal{J}$ sampled from $P' = \{\frac{1}{p_1}, \frac{1}{p_2}, ..., \frac{1}{p_N}\}$ follows the uniform distribution when $|\mathcal{J}| \to \infty$.*

**Theorem 3.1.** *When the batch size $|B| \to \infty$, items in the resampled set $\mathcal{R}_u$ with respect to the query $u$ based on sampling weight $w(i|u)$ follow the softmax distribution w.r.t. the query $u$.*

*Proof.* Assume $pop(i)$ denote the item frequency over whole item corpus, the probability of appearing in $\mathcal{R}_u$ can be obtained according to the multiplication rule in the following way:

$$
\begin{aligned}
P(i|i \in \mathcal{R}_u) &= P(i|i \in B) \cdot w(i|u) \\
&= pop(i) \cdot \frac{|B|}{|\mathcal{D}|} \cdot \frac{\exp(s(u,i) - \log pop(i))}{\sum_{j \in B} \exp(s(u,j) - \log pop(j))} \\
&= \frac{|B|}{|\mathcal{D}|} \cdot \frac{\exp(s(u,i))}{\sum_{j \in B} \frac{1}{pop(i)} \cdot \exp(s(u,j))} \\
&\approx \frac{|B|}{|\mathcal{D}|} \cdot \frac{\exp(s(u,i))}{\frac{|B|}{|\mathcal{D}|} \cdot \sum_{j \in \mathcal{I}} \exp(s(u,j))} = \frac{\exp(s(u,i))}{\sum_{j \in \mathcal{I}} \exp(s(u,j))}
\end{aligned}
$$

The approximately equal sign $\approx$ turns to the equal sign when $|B| \to \infty$. $\qquad\square$

According to the resampling approach, the sampled items follows the softmax distribution given the user query. Let's first review the expectation of gradient of the full softmax loss as follows:

$$\nabla \mathcal{L}_{\text{softmax}} = -\nabla s(u,i) + \sum_j^N \nabla s(u,j) = -\nabla s(u,i) + \mathbb{E}_{j \sim P^*(i|u)} \nabla s(u,j)$$

where $P^*(i|u) = \frac{\exp(s(u,i))}{\sum_{j \in \mathcal{I}} \exp(s(u,j))}$. Since the sampled items follows the softmax distribution, the estimated gradient with the item set $\mathcal{R}_u$ achieves an asymptotic-unbiased approximation of the expectation of gradient, i.e., $\sum_{i \in \mathcal{R}_u} \nabla s(u,i) \approx \mathbb{E}_{j \sim P^*(i|u)} \nabla s(u,j)$. Therefore, the objective function in a mini-batch has the following form:

$$\mathcal{L}_{\text{BIR}}(B, \Theta) = -\sum_{(u,i) \in B} \log \frac{\exp\left(s(u,i)\right)}{\sum_{j \in \mathcal{R}_u} \exp\left(s(u,j)\right)} \tag{3}$$

where $\mathcal{R}_u$ denotes the resampled item set with the same size as $B$. The Algorithm 1 sketches the pseudo-code for inbatch importance resampling. Compared with the naive inbatch sampling, BIR has the capability of distinguishing negative samples among different queries within inbatch items and capturing the dynamic changes of softmax, yielding an adaptive and flexible sampler to strengthen recommender retrievers. Note that the asymptotic-unbiased estimation is achieved when the batch size becomes extremely larger. A non-asymptotic bound can be attached with the following theorem.

---

**Algorithm 1:** Inbatch importance resampling (BIR)

---

**Input:** Training data $\mathcal{D} = \{(u,i)\}$, Number of epochs $T$
**Output:** Model parameters $\Theta$

1   Statistically get item popularity $P = \{pop(i) | \forall i \in \mathcal{I}\}$ from $\mathcal{D}$;
2   **for** $e = 1, 2, ..., T$ **do**
3     **for** *mini-batch* $B \in \mathcal{D}$ **do**
4       $U = \{u | (u,i) \in B\}$, $I = \{i | (u,i) \in B\}$;
5       Map query feature vectors into embeddings $E_U \in \mathbb{R}^{B \times k}$ based on $\phi_Q$;
6       Map item feature vectors into embeddings $E_I \in \mathbb{R}^{B \times k}$ based on $\phi_I$;
7       Calculate the matrix of similarity $S = E_U E_I^\top$;
8       **for** $u \in U$ **do**
9         Calculate the sampling probability $w(i|u) = \frac{\exp(s(u,i) - \log pop(i))}{\sum_{j \in I} \exp(s(u,j) - \log pop(j))}, \forall i \in I$;
10         Resample a set of items $\mathcal{R}_u$ based on $w(i|u)$;
11       Update model parameters based on the objective function, i.e., Eq (3);

---

**Theorem 3.2.** *Assum that the gradients of the logits $\nabla_\Theta s(u,i)$ have their coordinates bounded by $G$, the bias of $\nabla_\Theta \mathcal{L}_{BIR}$ satisfies:*

$$\mathbb{E}[\nabla_\Theta \mathcal{L}_{BIR}(u)] \leq \nabla_\Theta \mathcal{L}_{softmax}(u) + \mathbb{E}[U] \cdot \left( \frac{\sum_{i \in B} e^{s(u,i)} p(i) Z_{Bp} - Z_B^2}{|B| Z^3} + o\left(\frac{1}{|B|}\right) \right) +$$

$$\left( \frac{2G}{|B|} \cdot \frac{\max_{j,l \in B} |p(j) - p(l)| Z_{Bp} \cdot Z_B}{Z^2 + \sum_{i \in B} e^{s(u,i)} p(i) Z_{Bp}} + o\left(\frac{1}{|B|}\right) \right) \cdot \mathbf{1}$$

*where* $\mathbb{E}[U] = \sum_{i \in \mathcal{I}} \exp(s(u,i)) \nabla_\Theta s(u,i)$, $Z_{Bp} = \sum_{j \in B} \exp(s(u,j) - \log p(j))$, $Z_B = \sum_{j \in B} \exp(s(u,j))$, $Z = \sum_{i \in \mathcal{I}} \exp(s(u,i))$. *The probability $p(i)$ refers to the popularity $pop(i)$.*

This theorem can be proved according to Rawat's work [27] and more details can be attached in Appendix A.2. According to the theoretical results, with a larger batch size, the upper bound gets smaller, indicating that the estimated gradient is less biased. In terms of the popularity distribution, if the value of the popularity differs greatly, i.e., $\frac{\max pop(\cdot)}{\min pop(\cdot)}$ has a greater value, it would get a larger bias of the gradient.

### 3.2 Reusing Historical Informative Items with Cache-Augmented Resampling

Increasing the sample size is another crucial approach to controlling the bias and variance. Inspired by the recent work, MNS[34], which uses a mixture of inbatch negative samples and additional uniformly sampled negatives, we design a paradigm with cache-augmented samples, with the aim of reusing highly informative items in previous training epochs.

MNS feeds the model with additional uniformly sampled items, alleviating the Matthew effect to some extent. However, both the uniform distribution and popularity-based distribution are static from beginning to end, which may oversample the easy samples after several epochs. In order to reduce the bias of the query-based softmax during model training, we intuitively cache the items with averagely higher scores, depending on which historical harder negatives have the chance of being more trained.

More concretely, we count the occurrence number of times the items have been sampled into a vector $\boldsymbol{o} = \{o_1, o_2, ..., o_N\}$. If the item has been sampled for more times according to the aforementioned BIR, it has an averagely higher similar score among user queries, indicating that it has a higher probability of being hard negatives. The cache $\mathcal{C}$ is generated and updated by sampling items based on $o_i$, which serves as an additional candidate set to be resampled and augments the items from the original batch. Since the cache-augmented items follow a different distribution from the inbatch items, they can not share a unified normalization function. Thus, the items are resampled independently from the cache and the batch, and fit in two parts of loss as follows:

$$\mathcal{L}_{\chi IR}(B, \Theta) = -\lambda \sum_{(u,i) \in B} \log \frac{\exp(s(u,i))}{\sum_{j \in \mathcal{K}_u} \exp(s(u,j))} - (1-\lambda) \sum_{(u,i) \in B} \log \frac{\exp(s(u,i))}{\sum_{j \in \mathcal{R}_u} \exp(s(u,j))} \quad (4)$$

where $\lambda$ denotes a hyperparameter to control the influence of two separate parts. $\mathcal{K}_u$ denotes the item set resampled from the cache $\mathcal{C}$ based on the weight of $w_c = \frac{\exp(s(u,i)-\log q(i))}{\sum_{c \in \mathcal{C}} \exp(s(u,c)-\log q(c))}$, where $q(i) = pop(i)$. Note that the newly seen items are affected by a long-tailed distribution, we, therefore, correct the sampling bias for computing the normalized resampling weight. $\mathcal{R}_u$ refers to the resampled item set according to the inbatch importance resampling. In order to maintain a final resampled set with size $|B|$, we simply resample $\frac{1}{2}$ items from the cache and batch respectively, i.e., $|\mathcal{K}_u| = |\mathcal{R}_u| = \frac{|B|}{2}$. When newly resampled items are generated, the occurrence of items in the resampled set is updated. The cache-augmented item set offers more historical informative samples for training retrievers, being beneficial to getting rid of the sampling bias caused by the exposure bias. The Algorithm 2 details the procedure of the cache-augmented inbatch importance resampling.

---

**Algorithm 2:** Cache-Augmented Inbatch Importance Resampling ($\mathcal{X}$IR)

---

**Input:** Training data $\mathcal{D} = \{(u,i)\}$, Number of epochs $T$, Cache size $C$, Hyperparameter $\lambda$
**Output:** Model parameters $\Theta$

1   Initialize the occurrence vector $\boldsymbol{o} = \{0\}_{i=1}^{N}$ and the cache $\mathcal{C}$ with uniformly sampled items;
2   Statistically get item popularity $\mathcal{P} = \{pop(i)|\forall i \in \mathcal{I}\}$ from $\mathcal{D}$;
3   **for** $e = 1, 2, ..., T$ **do**
4      **for** *mini-batch* $B \in \mathcal{D}$ **do**
5         $U = \{u|(u,i) \in B\}, I = \{i|(u,i) \in B\}$;
6         Map user feature vectors into embeddings $E_U \in \mathbb{R}^{B \times k}$ based on $\phi_Q$;
7         Map item feature vectors into embeddings $E_I \in \mathbb{R}^{B \times k}$ based on $\phi_I$;
8         Map feature vectors for items in the cache $\mathcal{C}$ into embeddings $E_c \in \mathbb{R}^{C \times k}$ based on $\phi_I$;
9         Calculate the similarity matrix $S = E_U E_I^\top$ and $S' = E_U E_c^\top$;
10        **for** $u \in U$ **do**
11           Resample a set of items $\mathcal{R}_u$ based on $w(i|u) = \frac{\exp(s(u,i)-\log pop(i))}{\sum_{j \in I} \exp(s(u,j)-\log pop(j))}, \forall i \in I$;
12           Resample a set of items $\mathcal{K}_u$ based on $w_c(i|u) = \frac{\exp(s'(u,i)-\log q(i))}{\sum_{k \in \mathcal{C}} \exp(s'(u,k)-\log q(k))}, \forall i \in \mathcal{C}$ ;

        `// Update the cache and occurrence vector`
13        $\mathcal{O} = \{i|i \in \mathcal{K}_u \cup \mathcal{R}_u, \forall u \in U\}, o_i = o_i + \sum_{j \in \mathcal{O}} \mathbb{I}[j = i]$ ;
14        Sample $C$ items based on $w = o_i$ to update the cache $\mathcal{C}$;
15        Update model parameters based on the objective function, i.e., Eq (4);

---

### 3.3 Complexity Analysis

Compared with the inbatch sampling, BIR introduces additional computational cost for each training mini-batch of calculating the resampling weight (Line 9 in Alg 1) and resampling items according to the multinomial distribution (Line 10 in Alg 1), which has a time complexity of $\mathcal{O}(|B| \times |B|)$. Such time complexity is acceptable compared with the computational cost of embedding functions ($\mathcal{O}(\phi_Q)$ and $\mathcal{O}(\phi_I)$) and similarity matrix computation $\mathcal{O}(|B| \times |B| \times k)$. Regarding $\mathcal{X}$IR, it feeds the items of the cache into the item tower (Line 8 in Alg 2), taking the additional cost of embedding them. The time complexity of resampling is the same as that of BIR. Updating the cache and occurrences (Line 13-14 in Alg 2) takes $\mathcal{O}(N)$ time complexity, where $N$ is the number of items. BIR introduces additional $\mathcal{O}(|B| \times |B|)$ space to compute the similarity scores in a mini-batch, while $\mathcal{X}$IR further requires a space complexity of $\mathcal{O}(|\mathcal{C}| + N)$ to save the cache and occurrence vector.

## 4 Experiments

The proposed BIR and $\mathcal{X}$IR are then evaluated on five public real-world datasets in recommender retrievers to answer the following questions: (1) Can the inbatch importance resampling outperforms competitive strategies in inbatch sampling? (2) How do the cache-augmented items improve the performance of recommender retrievers?

## 4.1 Experimental Settings

### 4.1.1 Dataset

The experiments are conducted on the **Gowalla**, **Amazon**, **Ta-feng**, **Echonest** and **Tmall** datasets [2], as summarized in Table 1. The rated books in Amazon dataset with higher ratings than average ratings are considered as positive ones, while the interactions in other datasets are regarded as positive ones. 80% of items for a user are sampled for training and the other 20% of items constitute the test set. The performance of recommender retrievers is evaluated on the NDCG (Normalization Discounted Cumulative Gain) and RECALL at a cutoff of 10 by default.

Table 1: Statistics of Datasets

| Dataset | #User | #Item | #Interaction | Density |
|---------|-------|-------|--------------|---------|
| Gowalla | 29,858 | 40,988 | 1,027,464 | 8.40e-4 |
| Amazon | 130,380 | 128,939 | 2,415,650 | 1.44e-4 |
| Ta-feng | 19,451 | 10,480 | 630,767 | 3.09e-4 |
| Echonest | 217,966 | 60,654 | 4,422,471 | 3.35e-4 |
| Tmall | 125,553 | 58,058 | 2,064,290 | 2.83e-4 |

### 4.1.2 Baselines

In this work, we investigate the inbatch sampling for retrievers and we compare the proposed BIR and $\chi$IR [3] with competitive inbatch-sampling-based methods. All the methods attempt to minimize the log sampled softmax loss (i.e., Eq (2), namely SSL), varying in the selected negatives and the proposal distributions. (1) **SSL** utilizes the inbatch samples as negatives without the bias correction. (2) **SSL-Pop** uses the inbatch items and adopts the item popularity as the proposal distribution. (3) **MNS** [34] refers to the mixed negative sampling, which uses a mixture of batch and uniformly sampled negatives. We set the number of uniform samples $B'$ to be the same as the batch size. (4) **G-Tower** [35] corrects the selection bias in batch sampling by streaming frequency estimation. It records the last number of steps for each item in fixed-length hash arrays and reduces hash collision errors by using multiple hash arrays. In this way, the frequency is estimated as the reciprocal of the average number of steps. We use 5 different hash arrays to estimate.

### 4.1.3 Implementation Details

The proposed methods, including baseline methods, are implemented with the PyTorch learning framework in a Linux operating system and a Tesla V100 GPU. The Adam optimizer is utilized with a discount of 0.95 on learning rate at every 5 epochs. We use the ID features for users and items by default and the embedding size is set as 32. For each dataset, the batch size is fixed to 2048 and the learning rate is set to 0.001, where each dataset is training with 100 epochs. The coefficient of the $l_2$-regularization is tuned over $\{1e-4,, 1e-5, 1e-6\}$. The item popularity is calculated by the normalized item frequency, i.e., $p(i) = \frac{f_i}{\sum_j f_j}$ statistically summarized from training data. The hyperparameter $\lambda$ for $\chi$IR is tuned over $\{0.0, 0.2, 0.5, 0.8, 1.0\}$

## 4.2 Performance Comparisons with Baseline Methods

Each method is run for 5 times with different random seeds and we report the average scores with their standard deviations in Table 2. The proposed BIR and $\chi$IR considerably outperform the competitive baselines. BIR achieves a relative 2.47%, 6.57%, 6.58%, 2.04% and 5.17% improvements of NDCG@10 on the five datasets respectively while the relative improvements for $\chi$IR are 3.81%, 12.23%, 15.24%, 11.73% and 17.12%. Both BIR and $\chi$IR show superior performances, indicating the efficient and preferable utilization of the inbatch samples. In addition to correcting the selection bias by the item popularity, BIR and $\chi$IR can dynamically select more informative negative samples for different queries within the mini-batch, contributing to better model convergence. According to the importance resampling paradigm, the items with higher similarity scores with the corrected

---

[2]Datasets: https://recbole.io/cn/dataset_list.html

[3]Implementation for BIR and $\chi$IR: https://github.com/HERECJ/XIR

Table 2: Comparison with Baselines. $\delta = 1e - 4$

| Method | Gowalla | Amazon | Ta-feng | Echonest | Tmall |
|---|---|---|---|---|---|
| | NDCG@10 | NDCG@10 | NDCG@10 | NDCG@10 | NDCG@10 |
| SSL | $0.1381\pm5.5\delta$ | $0.0718\pm3.7\delta$ | $0.0437\pm9.4\delta$ | $0.1436\pm3.1\delta$ | $0.0340\pm3.1\delta$ |
| SSL-Pop | $0.1479\pm9.1\delta$ | $\underline{0.0764}\pm3.2\delta$ | $0.0625\pm1.5\delta$ | $0.1609\pm1.9\delta$ | $0.0547\pm5.6\delta$ |
| MNS | $0.1486\pm8.9\delta$ | $0.0781\pm4.9\delta$ | $\underline{0.0634}\pm1.4\delta$ | $\underline{0.1648}\pm2.5\delta$ | $\underline{0.0561}\pm4.9\delta$ |
| G-Tower | $\underline{0.1500}\pm8.6\delta$ | $\underline{0.0764}\pm4.0\delta$ | $0.0496\pm8.7\delta$ | $0.1600\pm5.1\delta$ | $0.0429\pm2.1\delta$ |
| BIR | $0.1523\pm7.9\delta$ | $0.0833\pm2.7\delta$ | $0.0675\pm1.3\delta$ | $0.1682\pm3.4\delta$ | $0.0590\pm3.2\delta$ |
| $\mathcal{X}$IR | $\mathbf{0.1543}\pm1.1\delta$ | $\mathbf{0.0877}\pm3.4\delta$ | $\mathbf{0.0730}\pm1.1\delta$ | $\mathbf{0.1842}\pm2.4\delta$ | $\mathbf{0.0658}\pm2.2\delta$ |
| | RECALL@10 | RECALL@10 | RECALL@10 | RECALL@10 | RECALL@10 |
| SSL | $0.1105\pm4.1\delta$ | $0.0753\pm4.5\delta$ | $0.0375\pm7.5\delta$ | $0.1350\pm3.6\delta$ | $0.0354\pm3.6\delta$ |
| SSL-Pop | $0.1124\pm4.4\delta$ | $0.0777\pm4.3\delta$ | $0.0506\pm9.2\delta$ | $0.1502\pm2.4\delta$ | $0.0544\pm6.1\delta$ |
| MNS | $0.1130\pm4.2\delta$ | $0.0796\pm6.1\delta$ | $\underline{0.0514}\pm8.5\delta$ | $\underline{0.1537}\pm2.9\delta$ | $\underline{0.0558}\pm5.4\delta$ |
| G-Tower | $\underline{0.1176}\pm6.8\delta$ | $\underline{0.0798}\pm5.8\delta$ | $0.0415\pm7.2\delta$ | $0.1504\pm5.1\delta$ | $0.0440\pm1.8\delta$ |
| BIR | $0.1157\pm4.6\delta$ | $0.0848\pm3.5\delta$ | $0.0549\pm8.6\delta$ | $0.1568\pm3.4\delta$ | $0.0588\pm3.1\delta$ |
| $\mathcal{X}$IR | $\mathbf{0.1169}\pm6.1\delta$ | $\mathbf{0.0895}\pm4.1\delta$ | $\mathbf{0.0589}\pm6.1\delta$ | $\mathbf{0.1703}\pm2.5\delta$ | $\mathbf{0.0651}\pm2.8\delta$ |

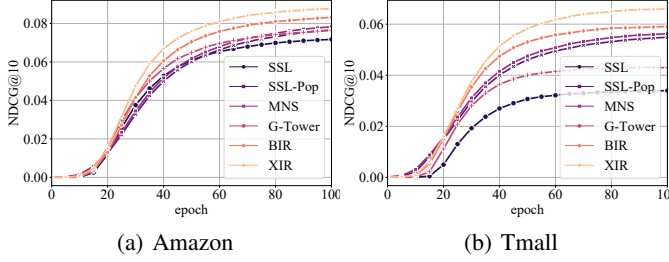

(a) Amazon     (b) Tmall

Figure 1: NDCG@10 vs. number of epochs on two datasets

Figure 2: Occurrence of items. Items are sorted in descending order of popularity

sampling bias have a higher probability of being resampled, where a more accurate estimation of the query-dependent softmax is achieved. In addition, we have the following findings.

*Finding 1:* Reducing the selection bias within the mini-batch by item popularity is beneficial to reducing the approximation bias from the softmax distribution. SSL, which does not adjust the proposal distribution, performs the worst among the baseline methods in terms of both NDCG and RECALL. Thus, such a method raises inconsistency between the real sampling distribution and the proposal distribution, resulting in poor performance of the sampled softmax loss.

*Finding 2:* Introducing negative items outside of the mini-batch expands the candidate set, facilitating the learning of retrievers. The result, that MNS outperforms SSL-Pop with an average 1.84% relative improvement of NDCG@10 and $\mathcal{X}$IR outperforms BIR with a 7.12% improvement, demonstrates that the model has a better resolution towards long-tailed items with the additional sampled items.

*Finding 3:* Caching the frequently sampled items helps improve the retriever performance, indicating that the frequently sampled items are more informative than uniformly sampled items. $\mathcal{X}$IR significantly improves the performance of BIR, where historical informative items have a tendency to remain in the cache with more training.

Furthermore, we investigate the recommendation performance during training epochs with different inbatch-sampling-based methods to verify the effectiveness of the BIR and $\mathcal{X}$IR. The changing curve of NDCG@10 on the Amazon and Tmall datasets are shown in Figure 1. Both BIR and $\mathcal{X}$IR contribute to faster convergence compared to the baselines. The model's NDCG quickly reaches a relatively high level under BIR and $\mathcal{X}$IR, particularly on the Amazon and Tmall datasets, implying that more informative items are selected for training.

## 4.3 Item Distribution in Cache

In order to intuitively figure out the item distribution in the cache, experiments are conducted to summarize the occurrence vector, since the cache is updated by sampling based on it. We summarize the occurrence after 100 epochs on the Gowalla dataset and normalize the item occurrence, as shown in Figure 2, where items are sorted by their popularity in descending order.

The distribution of items in the cache performs significantly different from the popularity-based distribution. The huge bias from the popularity-based distribution suggests that the cache can alleviate the long-tailed items becoming more popular according to the corrected sampling probability, i.e., the popularity. Moreover, the unpopular items have a greater chance of being sampled, whereas more informative items are encouraged to be more sampled, rather than treating full items as equivalent.

## 4.4 Adaptive Query-Dependent Estimation

Compared with MNS and G-Tower, where the items within the mini-batch are shared between user queries, the proposed BIR selects different samples, being able to approximate the softmax distributions for different queries and capture the dynamic changing of softmax distribution during the training process. In this chapter, we conduct experiments by resampling smaller numbers of items from the mini-batch and compare the performances with SSL-Pop. We report the relative improvements in terms of NDCG@10 and RECALL@10 in Figure 3(a) and 3(b).

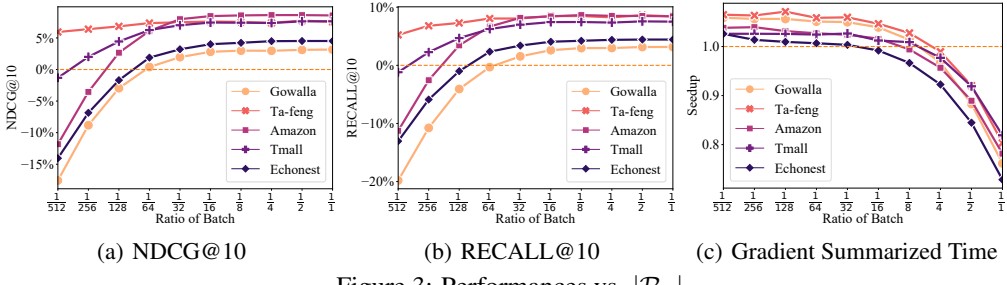

Figure 3: Performances vs. $|\mathcal{R}_u|$

From the figures, resampling nearly $\frac{1}{64}$ of items in the batch can achieve comparable performances of SSL-Pop with full batch items. Specifically, for Tmall, Amazon and Ta-feng datasets, it takes fewer samples to surpass SSL-Pop. This demonstrates the superiority of distinguished negative samples over different user queries, which performs a query-dependent approximation for softmax distribution and benefits the model training.

This series of experiments provides the potential for more efficient training with much fewer samples. Thus, we further compare the running time of summarizing the gradient under various sizes of the resampled item set in Figure 3(c). We run for 10 epochs for each setting and report the average time compared with SSL-Pop. Since data IO and feature embedding take considerably computational cost, we just compare the calculation on the gradient. As fewer items are selected, the summarized gradient receives fewer calculations.

## 4.5 Introducing Features into Item Tower

In online recommender retrievers, two tower models with deep networks offer the capacity of capturing complicated features. Thus, we further encode the categorical features *SellerID* and *CateID* in the Tmall dataset and build the item tower as two fully-connected layers with 32 and 16 units, using the Relu as the active function. As shown in Table 4, BIR and $\chi$IR outperform SSL-Pop, achieving a relative 3.06% and 4.87% improvements in terms of NDCG@10 respectively. This indicates the effectiveness of the proposed importance resampling based methods given more features, implying the possibility of deploying deeper retrieval towers.

| | N@10 | R@10 | N@20 | R@20 |
|---|---|---|---|---|
| SSL | 0.0311 | 0.0329 | 0.0524 | 0.0568 |
| SSL-Pop | 0.0720 | 0.0712 | 0.0999 | 0.1022 |
| BIR | 0.0742 | 0.0735 | 0.1032 | 0.1057 |
| $\chi$IR | 0.0755 | 0.0750 | 0.1067 | 0.1099 |

Figure 4: Performances on Tmall

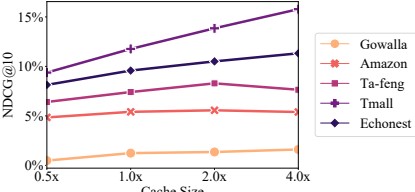

Figure 5: Performances vs. cache size

### 4.6 Effect of Cache Size

To figure out the influence of the cache size, we conduct experiments with various sizes, including 0.5x, 1x, 2x and 4x of the batch size (2048). We report the relative improvements compared with BIR in Figure 5. As the batch size gets larger, more items participate in the approximation of softmax, achieving a more accurate estimation and a better performance of recommendation quality.

### 4.7 Effect of $\lambda$

Since the coefficient plays a vital role in Eq (4), we conduct experiments by varying the value of $\lambda$ to figure out the influence of this hyperparameter. Experiments are conducted on all five datasets with $\chi$IR, as shown in Figure 6.

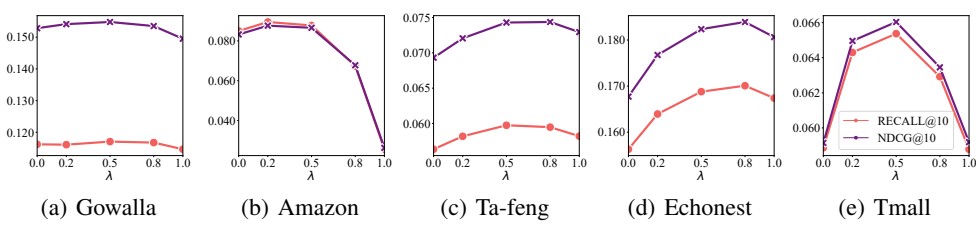

(a) Gowalla    (b) Amazon    (c) Ta-feng    (d) Echonest    (e) Tmall

Figure 6: Effect of $\lambda$

As shown in the figure, both the NDCG@10 and RECALL@10 have the same changing curve under different $\lambda$. A smaller value of $\lambda$ indicates the inbatch items contributes more gradient, whereas a lager value corresponds with that the cache plays a more important role. When $\lambda = 0.5$, the cached items and inbatch items contribute equally to convergence. For further study, we may link the resampled ratio with $\lambda$, instead of resampling half of the items from the cache and the batch.

## 5 Conclusion

In this paper, we propose a cache-augmented inbatch importance resampling for training recommender retrievers, which achieves a more accurate estimation of softmax for different queries using the inbatch items. The proposed BIR resamples items based on the similarity scores with popularity-based debias and thus different queries are trained with different sampled items, achieving a more accurate estimation of the softmax distribution. $\chi$IR is further proposed to encourage historical informative items to be more trained, which augments the inbatch items and thus helps retriever training. Experiments are finally conducted on five real-world datasets to validate the effectiveness of the proposed BIR and $\chi$IR.

## Acknowledgments and Disclosure of Funding

The work is supported by the National Natural Science Foundation of China (No. 61976198, 61972069, 61836007 and 61832017), and Shenzhen Municipal Science and Technology R&D Funding Basic Research Program (JCYJ20210324133607021).

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
