In the appendix, we first give the proofs of Lemma 3.1, Theorem 3.1 and Theorem 3.2 in section A. Then, we provide more empirical results for ablation studies.

## A   Theoretical Analysis

**Lemma A.1.** *(Lemma 3.1) Assume items follow a distribution $P = \{p_1, p_2, ..., p_N\}$, the item set $\mathcal{J}$ sampled from $P' = \{\frac{1}{p_1}, \frac{1}{p_2}, ..., \frac{1}{p_N}\}$ follows the uniform distribution when $|\mathcal{J}| \to \infty$. That is, for any set of items $\mathcal{A} \subseteq \mathcal{I}$, assuming items appear $K_{\mathcal{A}}$ times in $\mathcal{J}$, when $|\mathcal{J}| \to \infty$, $\frac{K_{\mathcal{A}}}{|\mathcal{J}|} = \frac{|\mathcal{A}|}{|\mathcal{I}|}$.*

*Proof.*

$$\frac{K_{\mathcal{A}}}{|\mathcal{J}|} \approx \sum_{i \in \mathcal{J} \cap \mathcal{A}} \frac{1}{p_i} = \frac{1}{|\mathcal{J}|} \sum_{i \in \mathcal{J}} \mathbb{I}[i \in \mathcal{A}] \cdot \frac{1}{p_i}$$

$$\approx \mathbb{E}_{i \sim P} \mathbb{I}[i \in \mathcal{A}] \cdot \frac{1}{p_i}$$

$$= \frac{1}{|\mathcal{I}|} \sum_{i \in \mathcal{I}} p_i \cdot \mathbb{I}[i \in \mathcal{A}] \cdot \frac{1}{p_i} = \frac{|\mathcal{A}|}{|\mathcal{I}|}$$

$\square$

**Theorem A.1.** *(Theorem 3.1) When the batch size $|B| \to \infty$, items in the resampled set $\mathcal{R}_u$ with respect to the query $u$ based on $w(i|u)$ follow the softmax distribution w.r.t. the query $u$.*

*Proof.* Assume $p(i)$ denote the item frequency over whole item corpus, the probability of appearing in $\mathcal{R}_u$ can be obtainded according to the multiplication rule in the following way:

$$P(i|i \in \mathcal{R}_u) = P(i|i \in B) \cdot w(i|u)$$

$$= p(i) \cdot \frac{|B|}{|\mathcal{D}|} \cdot \frac{\exp(s(u,i) - \log p(i))}{\sum_{j \in B} \exp(s(u,j) - \log p(j))}$$

$$= \frac{|B|}{|\mathcal{D}|} \cdot \frac{\exp(s(u,i))}{\sum_{j \in B} \frac{1}{p(i)} \cdot \exp(s(u,j))}$$

$$\approx \frac{|B|}{|\mathcal{D}|} \cdot \frac{\exp(s(u,i))}{\frac{|B|}{|\mathcal{D}|} \cdot \sum_{j \in \mathcal{I}} \exp(s(u,j))} = \frac{\exp(s(u,i))}{\sum_{j \in \mathcal{I}} \exp(s(u,j))}$$

The approximately equal sign $\approx$ turns to the equal sign when $|B| \to \infty$. $\square$

**Theorem A.2.** *(Theorem 3.2) Assum that the gradients of the logits $\nabla_{\Theta} s(u,i)$ have their coordinates bounded by $G$, the bias of $\nabla_{\Theta} \mathcal{L}_{BIR}$ satisfies:*

$$\mathbb{E}[\nabla_{\Theta} \mathcal{L}_{BIR}(u)] \leq \nabla_{\Theta} \mathcal{L}_{softmax}(u) + \mathbb{E}[U] \cdot \left( \frac{\sum_{i \in B} e^{s(u,i)} p(i) Z_{Bp} - Z_B^2}{|B| Z^3} + o\left(\frac{1}{|B|}\right) \right) +$$

$$\left( \frac{2G}{|B|} \cdot \frac{\max_{j,l \in B} |p(j) - p(l)| Z_{Bp} \cdot Z_B}{Z^2 + \sum_{i \in B} e^{s(u,i)} p(i) Z_{Bp}} + o\left(\frac{1}{|B|}\right) \right) \cdot \mathbf{1}$$

*where* $\mathbb{E}[U] = \sum_{i \in \mathcal{I}} \exp(s(u,i)) \nabla_{\Theta} s(u,i)$, $Z_{Bp} = \sum_{j \in B} \exp(s(u,j) - \log p(j))$, $Z_B = \sum_{j \in B} \exp(s(u,j))$, $Z = \sum_{i \in \mathcal{I}} \exp(s(u,i))$. *The probability $p(i)$ refers to the popularity $pop(i)$.*

*Proof.* The expectation of gradient in BIR with respect to the user $u$ follows as:

$$\mathbb{E}[\nabla_{\Theta} \mathcal{L}_{BIR}(u)] = -\nabla_{\Theta} s(u,i) + \mathbb{E}\left[ \frac{e^{s(u,i)} \cdot \nabla_{\Theta} s(u,i) + \sum_{j \in [m]} \left( \frac{e^{s(u,o_j)}}{m \cdot w(o_j|u)} \cdot \nabla_{\Theta} s(u,o_j) \right)}{e^{s(u,i)} + \sum_{j \in [m]} \frac{e^{s(u,o_j)}}{m \cdot w(o_j|u)}} \right]$$

where $w(j|u) = \frac{\exp(s(u,j) - \log p(j))}{\sum_{k \in B} \exp(s(u,k) - \log p(k))}$ with $p(i) = pop(i)$. $m$ denotes the number of resampled items and in this paper $m = |B|$. The sampled set is denoted as $\mathcal{R}_u = \{o_1, o_2, ..., o_m\}$, which is represented by $[m]$ for the sake of simplification.

Let us define random variables

$$U = e^{s(u,i)} \cdot \nabla_\Theta s(u,i) + \sum_{j \in [m]} \left( \frac{e^{s(u,o_j)}}{m \cdot w(o_j|u)} \cdot \nabla_\Theta s(u, o_j) \right)$$

and

$$V = e^{s(u,i)} + \sum_{j \in [m]} \frac{e^{s(u,o_j)}}{m \cdot w(o_j|u)}$$

Then we have

$$\mathbb{E}[U] = \sum_{i \in \mathcal{I}} e^{s(u,i)} \cdot \nabla_\Theta s(u,i) \text{ and } \mathbb{E}[V] = \sum_{i \in \mathcal{I}} e^{s(u,i)} = Z$$

According to the upper bound in Lemma A.2, we can obtain that

$$\mathbb{E}[\nabla_\Theta \mathcal{L}_{\text{BIR}}(u)] \leq -\nabla_\Theta s(u,i) + \mathbb{E}[U] \cdot \mathbb{E}\left[\frac{1}{V}\right] + \Delta_m$$

$$\Delta_m \triangleq \frac{1}{m} \mathbb{E}\left[ \sum_{k \in B} \frac{e^{s(u,k)} |\nabla_\Theta s(u,k)| \cdot |\frac{e^{s(u,o_m)}}{w(o_m|u)} - \frac{e^{s(u,o_k)}}{w(o_m|k)}|}{\left( e^{s(u,i)} + \sum_{j \in [m-1]} \frac{e^{s(u,o_j)}}{mw(o_j|u)} \right)^2} \right]$$

Then, by apply Lemma A.3, we can get the following

$$\mathbb{E}[\nabla_\Theta \mathcal{L}_{\text{BIR}}(u)] \leq -\nabla_\Theta s(u,i) + \frac{\mathbb{E}[U]}{\mathbb{E}[V]} + \mathbb{E}[U] \cdot \left( \frac{\sum_{k \in B} \frac{e^{2s(u,k)}}{w(k|u)} - \left( \sum_{k \in B} e^{s(u,k)} \right)^2}{mZ^3} + o\left(\frac{1}{m}\right) \right) + \Delta_m$$

$$= \nabla_\Theta \mathcal{L}_{\text{softmax}}(u) + \mathbb{E}[U] \cdot \left( \frac{\sum_{k \in B} \frac{e^{2s(u,k)}}{w(k|u)} - \left( \sum_{k \in B} e^{s(u,k)} \right)^2}{mZ^3} + o\left(\frac{1}{m}\right) \right) + \Delta_m$$

After that, we employ Lemma A.5 and get the upper bound as

$$\mathbb{E}[\nabla_\Theta \mathcal{L}_{\text{BIR}}(u)] \leq \nabla_\Theta \mathcal{L}_{\text{softmax}}(u) + \mathbb{E}[U] \cdot \left( \frac{\sum_{k \in B} \frac{e^{2s(u,k)}}{w(k|u)} - \left( \sum_{k \in B} e^{s(u,k)} \right)^2}{mZ^3} + o\left(\frac{1}{m}\right) \right) +$$

$$\left( \frac{2G}{m} \frac{\max_{j,l \in B} |\frac{e^{s(u,j)}}{w(j|u)} - \frac{e^{s(u,l)}}{w(l|k)}| \cdot \sum_{j \in B} e^{s(u,j)}}{Z^2 + \sum_{j \in B} \frac{e^{2s(u,j)}}{w(j|u)}} + o\left(\frac{1}{m}\right) \right) \cdot \mathbf{1}$$

Let us take into the specific value of $w(i|u)$ and obtain the following formula

$$\frac{e^{s(u,i)}}{w(i|u)} = \frac{e^{s(u,i)} \cdot \sum_{j \in B} e^{s(u,j)}/p(j)}{e^{s(u,i)}/p(i)} = p(i) \cdot \sum_{j \in B} e^{s(u,j)}/p(j) = p(i) Z_{Bp},$$

$$\frac{e^{2s(u,i)}}{w(i|u)} = p(i) \cdot e^{s(u,i)} \cdot Z_{Bp}, \quad \sum_{i \in B} e^{s(u,i)} = Z_B,$$

$$\max_{j,l \in B} |\frac{e^{s(u,j)}}{w(j|u)} - \frac{e^{s(u,l)}}{w(l|k)}| = \max_{j,l \in B} |p(j)Z_{Bp} - p(l)Z_{Bp}| = \max_{j,l \in B} |p(j) - p(l)| Z_{Bp}$$

Thus, we obtain the final result as

$$\mathbb{E}[\nabla_\Theta \mathcal{L}_{\text{BIR}}(u)] \leq \nabla_\Theta \mathcal{L}_{\text{softmax}}(u) + \mathbb{E}[U] \cdot \left( \frac{Z_{Bp} \cdot \sum_{k \in B} p(k) \cdot e^{s(u,k)} - (Z_B)^2}{mZ^3} + o\left(\frac{1}{m}\right) \right) +$$

$$\left( \frac{2G}{m} \frac{\max_{j,l \in B} |p(j) - p(l)| Z_{Bp} \cdot Z_B}{Z^2 + \sum_{i \in B} p(i) \cdot e^{s(u,i)} \cdot Z_{Bp}} + o\left(\frac{1}{m}\right) \right) \cdot \mathbf{1}$$

Since $m = |B|$, the upper bound becomes

$$\mathbb{E}[\nabla_\Theta \mathcal{L}_{\text{BIR}}(u)] \leq \nabla_\Theta \mathcal{L}_{\text{softmax}}(u) + \mathbb{E}[U] \cdot \left( \frac{Z_{Bp} \cdot \sum_{k \in B} p(k) \cdot e^{s(u,k)} - (Z_B)^2}{|B| Z^3} + o\left(\frac{1}{|B|}\right) \right) +$$

$$\left( \frac{2G}{|B|} \frac{\max_{j,l \in B} |p(j) - p(l)| Z_{Bp} \cdot Z_B}{Z^2 + \sum_{i \in B} p(i) \cdot e^{s(u,i)} \cdot Z_{Bp}} + o\left(\frac{1}{|B|}\right) \right) \cdot \mathbf{1}$$

$\square$

We can observe that the bias is correlated to the batch size (i.e., $|B|$) and the popularity distribution within the mini-batch (i.e., $\max_{j,l \in B} |p(j) - p(l)|$ ).

**Lemma A.2.** *Assume $\mathcal{R}_u = \{o_1, ..., o_m\} \subset B$ be $m$ i.i.d. drawn according to the sampling distribution $w(\cdot|u)$. Then, the ratio appearing in the gradient estimation based on the sampled softmax approach satisfies:*

$$\frac{e^{s(u,i)} \cdot \nabla_\Theta s(u,i)}{\sum_{j \in \mathcal{I}} e^{s(u,j)}} \cdot \sum_{k \in B} \frac{e^{s(u,k)} \cdot \nabla_\Theta s(u,k)}{e^{s(u,i)} + \frac{m-1}{m} \sum_{j \in B} e^{s(u,j)} + \frac{e^{s(u,k)}}{mw(k|u)}} \leq$$

$$\mathbb{E}\left[ \frac{e^{s(u,i)} \cdot \nabla_\Theta s(u,i) + \sum_{j \in [m]} \left( \frac{e^{s(u,o_j)}}{mw(o_j|u)} \cdot \nabla_\Theta s(u,o_j) \right)}{e^{s(u,i)} + \sum_{j \in [m]} \frac{e^{s(u,o_j)}}{mw(o_j|u)}} \right] \leq$$

$$\left( \sum_{k \in \mathcal{I}} e^{s(u,k)} \cdot \nabla_\Theta s(u,k) \right) \cdot \mathbb{E}\left[ \frac{1}{e^{s(u,i)} + \sum_{j \in [m]} \frac{e^{s(u,o_j)}}{mw(o_j|u)}} \right] + \Delta_m$$

*where*

$$\Delta_m \triangleq \frac{1}{m} \mathbb{E}\left[ \sum_{k \in B} \frac{e^{s(u,k)} |\nabla_\Theta s(u,k)| \cdot |\frac{e^{s(u,o_m)}}{w(o_m|u)} - \frac{e^{s(u,o_k)}}{w(o_m|k)}|}{\left( e^{s(u,i)} + \sum_{j \in [m-1]} \frac{e^{s(u,o_j)}}{mw(o_j|u)} \right)^2} \right]$$

*Proof.*

$$\mathbb{E}\left[ \frac{e^{s(u,i)} \cdot \nabla_\Theta s(u,i) + \sum_{j \in [m]} \left( \frac{e^{s(u,o_j)}}{mw(o_j|u)} \cdot \nabla_\Theta s(u,o_j) \right)}{e^{s(u,i)} + \sum_{j \in [m]} \frac{e^{s(u,o_j)}}{mw(o_j|u)}} \right]$$

$$= \mathbb{E}\left[ \frac{e^{s(u,i)} \cdot \nabla_\Theta s(u,i)}{e^{s(u,i)} + \sum_{j \in [m]} \frac{e^{s(u,o_j)}}{mw(o_j|u)}} \right] + \sum_{j \in [m]} \mathbb{E}\left[ \frac{\frac{e^{s(u,o_j)}}{mw(o_j|u)} \cdot \nabla_\Theta s(u,o_j)}{e^{s(u,i)} + \sum_{j \in [m]} \frac{e^{s(u,o_j)}}{mw(o_j|u)}} \right]$$

$$= \mathbb{E}\left[ \frac{e^{s(u,i)} \cdot \nabla_\Theta s(u,i)}{e^{s(u,i)} + \sum_{j \in [m]} \frac{e^{s(u,o_j)}}{mw(o_j|u)}} \right] + m\mathbb{E}\left[ \frac{\frac{e^{s(u,o_j)}}{mw(o_j|u)} \cdot \nabla_\Theta s(u,o_j)}{e^{s(u,i)} + \sum_{j \in [m]} \frac{e^{s(u,o_j)}}{mw(o_j|u)}} \right]$$

For $1 \leq l \leq m$, we define $S_l = \sum_{j \in [l]} \frac{e^{s(u,o_j)}}{w(o_j|u)}$,

$$\mathbb{E}\left[\frac{\frac{e^{s(u,o_j)}}{mw(o_j|u)} \cdot \nabla_\Theta s(u,o_j)}{e^{s(u,i)} + \sum_{j \in [m]} \frac{e^{s(u,o_j)}}{mw(o_j|u)}}\right]$$

$$=\mathbb{E}\left[\mathbb{E}\left[\frac{\frac{e^{s(u,o_j)}}{mw(o_j|u)} \cdot \nabla_\Theta s(u,o_j)}{e^{s(u,i)} + \frac{S_{m-1}}{m} + \frac{e^{s(u,o_m)}}{mw(o_m|u)}}\right]\Bigg| S_{m-1}\right]$$

$$=\mathbb{E}\left[\sum_{j \in B} w(k|u) \cdot \frac{\frac{e^{s(u,k)}}{mw(k|u)} \cdot \nabla_\Theta s(u,k)}{e^{s(u,i)} + \frac{S_{m-1}}{m} + \frac{e^{s(u,k)}}{mw(k|u)}}\right]$$

$$=\frac{1}{m}\sum_{j \in B}\mathbb{E}\left[\frac{e^{s(u,k)} \cdot \nabla_\Theta s(u,k)}{e^{s(u,i)} + \frac{S_{m-1}}{m} + \frac{e^{s(u,k)}}{mw(k|u)}}\right]$$

$$=\frac{1}{m}\sum_{j \in B}\mathbb{E}\left[\frac{e^{s(u,k)} \cdot \nabla_\Theta s(u,k)}{e^{s(u,i)} + \frac{S_{m-1}}{m} + \frac{e^{s(u,k)}}{mw(k|u)}} - \frac{e^{s(u,k)} \cdot \nabla_\Theta s(u,k)}{e^{s(u,i)} + \frac{S_m}{m}} + \frac{e^{s(u,k)} \cdot \nabla_\Theta s(u,k)}{e^{s(u,i)} + \frac{S_m}{m}}\right]$$

$$=\frac{1}{m}\sum_{j \in B}\mathbb{E}\left[\frac{e^{s(u,k)} \cdot \nabla_\Theta s(u,k)}{e^{s(u,i)} + \frac{S_{m-1}}{m} + \frac{e^{s(u,k)}}{mw(k|u)}} - \frac{e^{s(u,k)} \cdot \nabla_\Theta s(u,k)}{e^{s(u,i)} + \frac{S_m}{m}}\right] + \frac{1}{m}\sum_{j \in B}e^{s(u,j)}\nabla_\Theta s(u,j)\mathbb{E}\left[\frac{1}{e^{s(u,i)} + \frac{S_m}{m}}\right]$$

$$\leq\frac{1}{m^2}\mathbb{E}\left[\sum_{k \in B}\frac{e^{s(u,k)}|\nabla_\Theta s(u,k)| \cdot |\frac{e^{s(u,o_m)}}{w(o_m|u)} - \frac{e^{s(u,k)}}{w(k|u)}|}{\left(e^{s(u,i)} + \frac{S_{m-1}}{m} + \frac{e^{s(u,k)}}{mw(k|u)}\right) \cdot \left(e^{s(u,i)} + \frac{S_m}{m}\right)}\right] + \frac{1}{m}\sum_{j \in B}e^{s(u,j)}\nabla_\Theta s(u,j)\mathbb{E}\left[\frac{1}{e^{s(u,i)} + \frac{S_m}{m}}\right]$$

$$\leq\frac{1}{m^2}\mathbb{E}\left[\sum_{k \in B}\frac{e^{s(u,k)}|\nabla_\Theta s(u,k)| \cdot |\frac{e^{s(u,o_m)}}{w(o_m|u)} - \frac{e^{s(u,k)}}{w(k|u)}|}{\left(e^{s(u,i)} + \frac{S_m}{m}\right)^2}\right] + \frac{1}{m}\sum_{j \in B}e^{s(u,j)}\nabla_\Theta s(u,j)\mathbb{E}\left[\frac{1}{e^{s(u,i)} + \frac{S_m}{m}}\right]$$

After, we obtain the upper bound that

$$\mathbb{E}\left[\frac{\frac{e^{s(u,o_j)}}{mw(o_j|u)} \cdot \nabla_\Theta s(u,o_j)}{e^{s(u,i)} + \sum_{j \in [m]} \frac{e^{s(u,o_j)}}{mw(o_j|u)}}\right]$$

$$\leq e^{s(u,i)}\nabla_\Theta s(u,i)\mathbb{E}\left[\frac{1}{e^{s(u,i)} + \frac{S_m}{m}}\right] + \sum_{j \in B}e^{s(u,j)}\nabla_\Theta s(u,j)\mathbb{E}\left[\frac{1}{e^{s(u,i)} + \frac{S_m}{m}}\right] +$$

$$\frac{1}{m}\mathbb{E}\left[\sum_{k \in B}\frac{e^{s(u,k)}|\nabla_\Theta s(u,k)| \cdot |\frac{e^{s(u,o_m)}}{w(o_m|u)} - \frac{e^{s(u,k)}}{w(k|u)}|}{\left(e^{s(u,i)} + \frac{S_m}{m}\right)^2}\right]$$

$$leq \sum_{j \in \mathcal{I}}e^{s(u,j)}\nabla_\Theta s(u,j) \cdot \mathbb{E}\left[\frac{1}{e^{s(u,i)} + \frac{S_m}{m}}\right] + \frac{1}{m}\mathbb{E}\left[\sum_{k \in B}\frac{e^{s(u,k)}|\nabla_\Theta s(u,k)| \cdot |\frac{e^{s(u,o_m)}}{w(o_m|u)} - \frac{e^{s(u,k)}}{w(k|u)}|}{\left(e^{s(u,i)} + \frac{S_m}{m}\right)^2}\right]$$

$$=\sum_{j \in \mathcal{I}}e^{s(u,j)}\nabla_\Theta s(u,j) \cdot \mathbb{E}\left[\frac{1}{e^{s(u,i)} + \frac{S_m}{m}}\right] + \Delta_m$$

$\square$

**Lemma A.3.** *Consider the random variable,*

$$V = e^{s(u,i)} + \sum_{j \in [m]} \frac{e^{s(u,o_j)}}{m \cdot w(o_j|u)}$$

*we have*

$$\mathbb{E}\left[\frac{1}{V}\right] \leq \frac{1}{\mathbb{E}[V]} + \frac{\sum_{k \in B}\frac{e^{2s(u,k)}}{w(k|u)} - \left(\sum_{k \in B}e^{s(u,k)}\right)^2}{mZ^3} + o\left(\frac{1}{m}\right)$$

*Proof.* As provided by Lemma A.6 that $\frac{1}{\mathbb{E}[V]} \leq \mathbb{E}\left[\frac{1}{V}\right] \leq \mathbb{E}\left[\frac{1}{V}\right] + \frac{Var(V)}{\mathbb{E}[V]^3} + o\left(\frac{1}{m}\right)$,

$$\mathbb{E}[V^2] = e^{2s(u,i)} + \frac{e^{s(u,i)}}{m}\mathbb{E}\left[\sum_{j\in[m]} \frac{2e^{s(u,o_j)}}{w(o_j|u)}\right] + \frac{1}{m^2}\mathbb{E}\left[\sum_{j,lin[B]} \frac{e^{s(u,o_j)} + e^{s(u,o_l)}}{w(o_j|u)w(o_l|u)}\right]$$

$$= \mathbb{E}[V]^2 + \frac{1}{m}\sum_{j\in B} \frac{2e^{s(u,j)}}{w(j|u)} - \frac{1}{m}\sum_{j,l\in B} e^{s(u,j)} + e^{s(u,l)}$$

Then the variance is obtained

$$Var(V) = \mathbb{E}[V^2] - \mathbb{E}[V]^2 = \frac{1}{m}\sum_{j\in B} \frac{2e^{s(u,j)}}{w(j|u)} - \frac{1}{m}\sum_{j,l\in B} e^{s(u,j)} + e^{s(u,l)}$$

Thus,

$$\frac{1}{\mathbb{E}[V]} \leq \mathbb{E}\left[\frac{1}{V}\right] + \frac{\sum_{j\in B}\frac{2e^{s(u,j)}}{w(j|u)} - \sum_{j,l\in B}e^{s(u,j)} + e^{s(u,l)}}{mZ^3} + o\left(\frac{1}{m}\right)$$

$$= \mathbb{E}\left[\frac{1}{V}\right] + \frac{\sum_{j\in B}\frac{2e^{s(u,j)}}{w(j|u)} - \left(\sum_{j\in B}e^{s(u,j)}\right)^2}{mZ^3} + o\left(\frac{1}{m}\right)$$

$\square$

**Lemma A.4.** *Consider the random variable*

$$W = \left(e^{s(u,i)} + \frac{S_{m-1}}{m}\right)^2$$

*we have*

$$\mathbb{E}\left[\frac{1}{W}\right] = \frac{1}{Z^2 + \sum_{j\in B}\frac{e^{s(u,j)}}{w(j|u)}} + o\left(\frac{1}{m}\right)$$

*Proof.*

$$\mathbb{E}[W] = \mathbb{E}\left[\left(e^{s(u,i)} + \frac{S_{m-1}}{m}\right)^2\right]$$

$$= Z^2 + \sum_{j\in B}\frac{e^{2s(u,j)}}{w(j|u)} - \frac{2}{m}\cdot e^{s(u,i)}\sum_{j\in B}e^{s(u,j)} - \frac{1}{m}\sum_{j\in B}\frac{e^{2s(u,j)}}{w(j|u)} - \frac{3m-2}{m^2}\sum_{j,l\in B}e^{s(u,j)+s(u,l)}$$

$$= \widetilde{W}$$

According to Lemma A.6 $\mathbb{E}\left[\frac{1}{W}\right] \leq \frac{1}{\mathbb{E}[W]} + \frac{Var(W)}{e^{6s(u,i)}} = \frac{1}{\mathbb{E}[W]} + \mathcal{O}\left(\frac{1}{m}\right)$, we have

$$\mathbb{E}\left[\frac{1}{W}\right] \leq \frac{1}{\widetilde{W}} + \mathcal{O}\left(\frac{1}{m}\right) \leq \frac{1}{Z^2 + \sum_{j\in B}\frac{e^{s(u,j)}}{w(j|u)}} + \mathcal{O}\left(\frac{1}{m}\right)$$

$\square$

**Lemma A.5.** *For any model parameter $\Theta \in \{\Theta\}$, by assuming that the coordinates of the gradient vectors have the following bound*

$$\|\nabla_\Theta s(u,k)\|_\infty \leq G, \forall k \in \mathcal{I}$$

*Then, the variable $\Delta_m$ satisfies*

$$\Delta_m \triangleq \frac{1}{m}\mathbb{E}\left[\sum_{k\in B}\frac{e^{s(u,k)}|\nabla_\Theta s(u,k)|\cdot|\frac{e^{s(u,o_m)}}{w(o_m|u)} - \frac{e^{s(u,o_k)}}{w(o_k|u)}|}{\left(e^{s(u,i)} + \sum_{j\in[m-1]}\frac{e^{s(u,o_j)}}{mw(o_j|u)}\right)^2}\right]$$

$$\leq \left(\frac{2G}{m}\frac{\max_{j,l\in B}|\frac{e^{s(u,j)}}{w(j|u)} - \frac{e^{s(u,l)}}{w(l|k)}|\cdot\sum_{j\in B}e^{s(u,j)}}{Z^2 + \sum_{j\in B}\frac{e^{2s(u,j)}}{w(j|u)}} + o\left(\frac{1}{m}\right)\right)\cdot\mathbf{1}$$

*Proof.*

$$\Delta_m = \frac{1}{m}\mathbb{E}\left[\sum_{k\in B}\frac{e^{s(u,k)}|\nabla_\Theta s(u,k)|\cdot|\frac{e^{s(u,o_m)}}{w(o_m|u)}-\frac{e^{s(u,o_k)}}{w(o_k|u)}|}{\left(e^{s(u,i)}+\sum_{j\in[m-1]}\frac{e^{s(u,o_j)}}{mw(o_j|u)}\right)^2}\right]$$

$$\leq \frac{1}{m}\mathbb{E}\left[\sum_{k\in B}\frac{e^{s(u,k)}|\nabla_\Theta s(u,k)|\cdot\max_{j,l\in B}|\frac{e^{s(u,j)}}{w(o_j)}-\frac{e^{s(u,l)}}{w(l|u)}|}{\left(e^{s(u,i)}+\sum_{j\in[m-1]}\frac{e^{s(u,o_j)}}{mw(o_j|u)}\right)^2}\right]$$

$$\leq \frac{2\cdot\max_{j,l\in B}|\frac{e^{s(u,j)}}{w(o_j)}-\frac{e^{s(u,l)}}{w(l|u)}|}{m}\cdot\mathbb{E}\left[\frac{1}{\left(e^{s(u,i)}+\frac{S_{m-1}}{m}\right)^2}\right]\cdot\sum_{k\in B}e^{s(u,k)}|\nabla_\Theta s(u,k)|$$

$$\leq \frac{2\cdot G\cdot\max_{j,l\in B}|\frac{e^{s(u,j)}}{w(o_j)}-\frac{e^{s(u,l)}}{w(l|u)}|}{m}\cdot\mathbb{E}\left[\frac{1}{\left(e^{s(u,i)}+\frac{S_{m-1}}{m}\right)^2}\right]\cdot\left(\sum_{k\in B}e^{s(u,k)}\right)\cdot\mathbf{1}$$

By employing Lemma A.4, we have that

$$\Delta_m \leq \left(\frac{2G}{m}\frac{\max_{j,l\in B}|\frac{e^{s(u,j)}}{w(j|u)}-\frac{e^{s(u,l)}}{w(l|k)}|\cdot\sum_{j\in B}e^{s(u,j)}}{Z^2+\sum_{j\in B}\frac{e^{2s(u,j)}}{w(j|u)}}+o\left(\frac{1}{m}\right)\right)\cdot\mathbf{1}$$

$\square$

**Lemma A.6.** *For a positive random variable $V$ and $V \geq a > 0$, the expectation follows*

$$\frac{1}{\mathbb{E}[V]}\leq\mathbb{E}\left[\frac{1}{V}\right]\leq\mathbb{E}\left[\frac{1}{V}\right]+\frac{Var(V)}{a^3}$$

*and*

$$\frac{1}{\mathbb{E}[V]}\leq\mathbb{E}\left[\frac{1}{V}\right]\leq\mathbb{E}\left[\frac{1}{V}\right]+\frac{Var(V)}{\mathbb{E}[V]^3}+\frac{\mathbb{E}|V-\mathbb{E}[V]|^3}{a^4}$$

The proof can be attached according to Rawat's work [27].

# B  Experimental Supplements

## B.1  Varying batch sizes

In this chapter, we further explore BIR by taking a look at how NDCG@10 changes under different batch sizes, i.e., $|B|$. Experiments are conducted on the five datasets with SSL-Pop and BIR, as shown in Figure 7. The batch sizes range from 256 to 8192. We fix the learning epoch to 100, and tune the learning rate and the weight decay depending on NDCG@10 for different batch sizes.

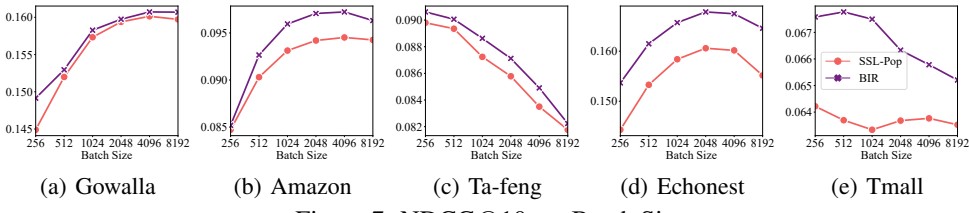

| (a) Gowalla | (b) Amazon | (c) Ta-feng | (d) Echonest | (e) Tmall |

Figure 7: NDCG@10 vs. Batch Size

Considering the real online scenarios with numerous features for both users and items, the batch size would be restricted to small values due to the limited computational resources. The performance under small batch size is more crucial for deployment. BIR consistently improves the performances even under smaller batch sizes, indicating the feasibility for online systems.