# OpenReview forum: "Cache-Augmented Inbatch Importance Resampling for Training Recommender Retriever"
_NeurIPS.cc/2022/Conference — NeurIPS 2022 Accept_

### Official Review · Reviewer_94bj · 2022-06-21

**Rating:** 7
**Confidence:** 5
**Soundness:** 3 good
**Presentation:** 3 good
**Contribution:** 3 good

**Summary:**

Large-scale two-tower recommendation systems are the norm in the industry. One of the biggest challanges in model training is the growing data set sizes. This even makes constructing the training set somewhat tricky--often pushing the modelers randomly sample their data set. This is sub-optimal. The paper proposes a new model training process that uses an in-batch importance sampling in distributed systems to train large-scale recommender systems. The proposed techniques have been viewed in two segments--1) importance sampling 2) cache-augmentation. The authors compare their methods with appropriate and standard baselines and analyze the parameters to inform future users.

**Questions:**

Along the lines of relating the literature to baselines and final learnings/findings:
What would be the recommendation of the authors to readers on which techniques to use under what conditions?


**Limitations:**

There could be more such discussion. The importance of sampling, particularly the one done through learned embeddings, can result in significant biases in the systems. Depending on the learned embeddings, any such bias could be heavily boosted in certain scenarios. On the other hand, the competition hypothesis such as popularity-based techniques are not necessarily better (which is shown) and potentially in many scenarios, they could be even worse. This is not a critique of the proposed technique but this aspect of the problem is not well studied in the paper.

On the other hand, I understand the fact that this itself could be a separate paper.  However, the paper proposes a weighting technique alternative to random sampling, this topic is at the heart of the problem.

**Strengths And Weaknesses:**

A) Strengths

-The paper touches on a simple but essential step of model training that is often neglected and, via a simple idea, dramatically improves model performance as well as provides the ability to increase model complexity or gain training time potentially.

-The authors demonstrate that a modern ML production system should think about every single step end-to-end holistically and use the power of systems and algorithms jointly to optimize the whole process.

-The paper is written well and rigorously explained, and enough technical details are provided. Given the space, the experimentation depth (both baseline comparison and analysis of algorithms) and findings/learnings are pretty detailed.

B) Weaknesses (I don't have any major concerns. Just nit-picking):

-There might be some weaknesses in the literature review and how they do relate to the baselines picked.

-Not enough discussion on how these techniques would provide biases and affect the populations/society.

-It might be good to discuss future work, particularly along the lines of candidate generation and retrieval for inference stages. I suspect several such learning could inform the research in this direction.

---

> ### Author Response · Authors · 2022-08-02
> **Reply to Reviewer 94bj**
>
> Thank you for your valuable reviews. Here are our responses to your comments.
> 1. [Picked Baselines]
>     > **Comment**: There might be some weaknesses in the literature review and how they do relate to the baselines picked.
>
>     This paper focuses on the inbatch sampling for training retrievers in real-world online systems, and we also discuss the various samplers in more general cases for recommender systems in section 2.2. The chosen baselines are proposed to conduct effective in-batch sampling strategies to enhance performance. All baseline methods use the other training items within the mini-batch as negatives to optimize the sampled softmax loss. SSL denotes the native sampled softmax loss while SSL-Pop corrects the sampling bias with the popularity weights. G-Tower estimates the item frequency under streaming data and MNS feeds the mini-batch with the additionally uniformly sampled items.
> 2. [Discussion about bias]
>     > **Comment**: the competition hypothesis such as popularity-based techniques are not necessarily better (which is shown) and potentially in many scenarios, they could be even worse. This is not a critique of the proposed technique but this aspect of the problem is not well studied in the paper.
>
>     The popularity-based technique does not always produce better results because it is dependent on whether the ultimate optimization goal is influenced by the popularity-based distribution of data or labels. For training retrievers with the sampled softmax, the training data within the mini-batch is skewed and thus the debias of the sample weights depending on the popularity-based distribution can keep the actual sampling distribution and sampling weight consistent. Thus,  as shown in the experimental results (SSL performs worst because it does not correct the bias of the inbatch sampling), the approaches considering the popularity-based distribution significantly improve performance.
>
> 3.  [Suitable Situations]
>     > **Comment**: What would be the recommendation of the authors to readers on which techniques to use under what conditions?]
>
>     In this paper, we analyze the inbatch sampling for training retrievers in nowadays online systems and suggest the importance-resampling-based method to sample query-dependent samples for better training.
>     In a broader sense, our proposed sampler is applicable to the algorithms for optimizing ranking metrics, such as NDCG and RECALL, in recommender systems and retrieval systems, especially when the amount of data is large, the features are numerous, and the model is complex.
>
> 4. [Future Work]
>     > **Comment**: It might be good to discuss future work, particularly along the lines of candidate generation and retrieval for inference stages.
>
>     Thank you for your advice on further discussing the retriever tasks. We will attempt to design a better theoretically provable cache to train the retrievers.

---

### Official Review · Reviewer_rEcV · 2022-07-10

**Rating:** 6
**Confidence:** 3
**Soundness:** 3 good
**Presentation:** 2 fair
**Contribution:** 2 fair

**Summary:**

The paper discusses the drawback of traditional negative sampling methods for the retrieval task for general recommender systems, and then proposes a new method which samples negative data points specifically for each user/query input, and illustrates why it could better approximate the gradient during model training. Experiments are provided to support the advantage of the proposed method.

**Questions:**

1. I want to understand the value proposition of this paper. Are we designing a practically useful algorithm or it's about some innovations behind the theorem? In its current shape, the paper seems a mixture of both. As a result, the level of contribution cannot be well estimated. For example, if it's about theoretical innovation, then the techniques behind the proof should be highlighted; if it's about practically useful algorithms, then I don't think there are enough material in the experiments and the discussion of real-world IR system.

2. The paper repeatedly mentioned about characteristics and contraints of modern hardware, which also guided the design of the new method. I suggest to explicitly talk about related details. Does it mean the new method is only useful for a particularly implemented system under certain architecture?

3. Discussion of the theorem. We do have the real-world data. Does the theorem lead to a significant difference under the usually used batch size?

4. It's now very clear about the usefulness of the cache. I feel the advantage is again a mixture of implementation and algorithm. I didn't find much discussion on this. It seems to me that the assumption is that getting the representation is slow.

**Limitations:**

Though not very explicitly mentioned, the new method still has bias for the estimate of the gradient. This can be considered as the limitation.

**Strengths And Weaknesses:**

Strength:

1. The problem itself is important. For extremely large scale retrieval system, the negative sampling is the key for training. The authors also capture the critical drawbacks of existing negative sampling method.

2. The bias reducing theorem is a highlight of the paper.


Weakness:

1. I also want to mention theorem 3.1. Thought I don't know how difficult it is, but a finite batch size theorem would make the paper much stronger because it would be far more practical. In reality, other factors could dominate the training performance which the benefit of the new method cannot be revealed when the batch size is not big enough.

2. For the experiments, one missing piece is the discussion of dataset sizes. The sizes are illustrated but are they large enough to fit the motivation behind the new method? Given the size, it seems naive global negative sampling can work.

---

> ### Author Response · Authors · 2022-08-02
> **Reply to Reviewer rEcV**
>
> Thank you for your valuable reviews. Here are our responses to your comments.
>
> 1. [Impact of Batch size]
>     > **Comment1**: but a finite batch size theorem would make the paper much stronger because it would be far more practical.
>
>     > **Comment2**: Discussion of the theorem. We do have the real-world data. Does the theorem lead to a significant difference under the usually used batch size?
>
>     a. Theoretical Support on Finite Batch Size
>
>     Theorem 3.1 states that the resampled items follow the softmax distribution given an infinite number of batch size. In Theorem 3.2, we examine the bias of the gradient to assess the effect of the finite batch size. The upper bound of the bias gets smaller as batch size increases, indicating that the estimated gradient is less biased.
>
>     b. Experimental Results
>
>     In Appendix B.3, we present the experimental results for the five datasets with varying batch sizes. BIR improves performance even with smaller batch sizes, indicating the feasibility of the proposed method for online systems.
>
> 2. [Size of dataset]
>     > **Comment**: For the experiments, one missing piece is the discussion of dataset sizes. The sizes are illustrated but are they large enough to fit the motivation behind the new method? Given the size, it seems naive global negative sampling can work.
>
>     The evaluated datasets, Amazon and Echonest, are relatively large among public recommendation datasets in terms of the corpus size and the number of interactions. In previous studies for the inbatch sampling for recommender retrievers, i.e., G-Tower[1] and MNS[2], they conduct experiments on private real-world datasets. We also anticipate the availability of much larger datasets for future research.
>
> 3. [Contribution - Algorithm or Theorem]
>     > **Comment**: I want to understand the value proposition of this paper. Are we designing a practically useful algorithm or it's about some innovations behind the theorem?
>
>     In this paper, we design a sampler with guaranteed theoretical provable and we follow the machine learning research paradigm to extensively verify the effectiveness from theoretical and experimental perspectives.
>
> 4. [Modern Hardware & Limited Architecture]
>     > **Comment**: The paper repeatedly mentioned about characteristics and contraints of modern hardware, which also guided the design of the new method. I suggest to explicitly talk about related details. Does it mean the new method is only useful for a particularly implemented system under certain architecture?
>
>     We motivate the research problem through the characteristics and constraints of modern hardware, but our algorithm can be suitable for modern computing architecture. Current models are trained on devices with limited resources, such as GPUs and TPUs. In the industrial recommendation scenarios with a great number of items, the training process is performed through mini-batches on the current computing architecture. There are numerous features and complex models in realistic recommenders, while the adaptive sampler is inefficient over the entire item set, relying heavily on the inference results from the complex models.
>
>     > **Comment**: It's now very clear about the usefulness of the cache. I feel the advantage is again a mixture of implementation and algorithm. I didn't find much discussion on this. It seems to me that the assumption is that getting the representation is slow.
>
>     Representation is a part of the model inference process, and inferring the entire item corpus is inefficient. The designed cache mechanism can achieve a compromise between global and inbatch data.
>
>     The cache is designed to reuse relatively informative samples from previous training epochs, with the probability of being cached based on the previous occurrence. As we can see in Figure 2, the occurrence distribution for generating the cache performs significantly differently than the item distribution (the popularity-based distribution), where the long-tailed items have a greater chance of being selected than previous native inbatch sampling.
>
> Reference
>
> [1]Yi X, Yang J, Hong L, et al. Sampling-bias-corrected neural modeling for large corpus item recommendations// RecSys 2019
>
> [2]Yang J, Yi X, Zhiyuan Cheng D, et al. Mixed negative sampling for learning two-tower neural networks in recommendations// WWW 2020 Companion

---

> ### Comment · Reviewer_rEcV · 2022-08-10
> **Rebuttal considered**
>
> Thanks for the rebuttal and appreciate the efforts. I think most of my comments/questions are addressed. Other reviews and related discussions also resolve my concern on the level of contributed. Changing the rating to 6.

---

### Official Review · Reviewer_mWFy · 2022-07-10

**Rating:** 6
**Confidence:** 5
**Soundness:** 2 fair
**Presentation:** 3 good
**Contribution:** 2 fair

**Summary:**

This work designs an inbatch negative sampling method for improving both effectiveness and efficiency of contrastive item recommendation task.

**Questions:**

1. Could you please compare baselines like DNS, LambdaFM, PRIS?
2. It should be essential capacity of negative sampling method to deal with false negative samples. Please justify how the proposed method can make it?
3. Please give more discussion on the superiority of the proposed method comparing with the other kinds of sampling method which does not attempt to sample from a softmax distribution in the introduction section.

**Limitations:**

Yes

**Strengths And Weaknesses:**

Strengths:
1. The proposed method is well formalized with theoretical support.
2. The presentation is overall good enough.
3. Experiments are conducted on large-scale datasets to evaluate the performance of the proposed approach.

Weakness:
1. The proposed method ignores the false negative sample issue if sampling from the full softmax distribution. An item with large similarity does not have to be a hard negative sample.
2. This work mainly focuses on how to improve the sample efficiency from softmax distribution. However, it does not present the superiority of the proposed method comparing to previous works that are not based on importance sampling from softmax distribution, for example, dynamic negative sampling (DNS, Weinan Zhang et al. SIGIR 2013), lambdaFM (CIKM 2016), PRIS (The WebConf 2020), AOBPR (WSDM 2014) etc. It deserves more words to compare with methods beyond estimating softmax distribution.
3. The baselines are not fully compared to validate the superiority of the proposed method. At least, the typical method PRIS should be a strong baseline which also estimates the softmax distribution with importance sampling method.

---

> ### Author Response · Authors · 2022-08-02
> **Reply to Reviewer mWFy**
>
> Thank you for your valuable reviews. Here are our responses to your comments.
>
> 1. [False negatives]
>     > **Comment1**: The proposed method ignores the false negative sample issue if sampling from the full softmax distribution. An item with large similarity does not have to be a hard negative sample.
>
>     > **Comment2**: It should be essential capacity of negative sampling method to deal with false negative samples. Please justify how the proposed method can make it?
>
>     False negative is indeed a challenging issue in negative sampling for recommender systems. In order to investigate the effect of false negatives in our samplers, we summarize the sampling probability of 'false negatives', which experimentally refer to items appearing in the testing data but not in the training data. Specifically, we calculate the average sampling probability in two ways. $$p_{global} = \frac{1}{|D_{test}|} \sum_{(u,j) \in D_{test}} p(j|u)$$ where $p(j|u) = \frac{\exp(s(u,j))}{\sum_{i\in \mathcal{I}} \exp(s(u,i))}$  denotes the softmax probability calculated by the well-trained model. This one describes the sampling probability over all the interacted items in the testing data. $$p_{user} = \frac{1}{M}\sum_{u} \sum_{(u,i) \in  D_{test}}p(j|u)$$ This average value represents the probability of sampling false negatives given a user.
>
>
>     | DataSet | $p_{global}$ | $p_{user}$ |
>     | :-----: | :----------: | :--------: |
>     | Gowalla |    0.00050   |   0.00345  |
>     | Amazon  |    0.00015   |   0.00050  |
>
>     As we can see in the table, the sampling probability for negative samples is relatively low. On the Gowalla dataset, the average probability of a negative sample being sampled is only 0.345% given a certain user. On the much larger dataset, Amazon, the probability of sampling false negatives decreases, indicating a lower likelihood of being sampled.
>
>
> 2. [Focus on sampling from a softmax distribution]
>     > **Comment1**: It deserves more words to compare with methods beyond estimating softmax distribution.
>
>     > **Comment2**: Please give more discussion on the superiority of the proposed method comparing with the other kinds of sampling method which does not attempt to sample from a softmax distribution in the introduction section.
>
>     There are various types of loss functions in general recommender systems and information retrieval, such as point-wise loss (binary cross-entropy loss), pair-wise loss (BPR loss), etc. However, recommender retrievers aim to recall a fraction of items for the subsequent stage, and are optimized under the log-softmax loss, which is more consistent with the rank-based metrics than the pair-wise and point-wise losses. When encountered with numerous candidate items, negative samplers provide an estimation of the original softmax distribution with a smaller number of items, significantly reducing time complexity. Thus, we concentrate primarily on samplers for estimating the softmax distribution.
>
> 3. [Comparison with Suggested Baselines]
>     > **Comment1**: At least, the typical method PRIS should be a strong baseline which also estimates the softmax distribution with importance sampling method.
>
>     > **Comment2**: Could you please compare baselines like DNS, LambdaFM, PRIS?
>
>     We perform experiments on the four datasets to compare with the suggested baselines, i.e., DNS, LambdaFM and PRIS. The three baseline samplers are designed to optimize the pair-wise loss. PRIS with uniform distribution and popularity-based distribution are denoted as PRIS(U) and PRIS(P), respectively. The batch size is set as 2048. To keep the sampled set the same size in this paper, each query is compared with 2048 sampled negatives. The learning rate is fixed as 0.01 and the weight decay is tuned over $\{0.01, 0.001, 0.0001, 0.00001\}$. The results (i.e., NDCG@10) are reported in the table below.
>
>     |          |  DNS   | LambdaFM | PRISU  | PRISP  | **BIR** | **XIR** |
>     |:--------:|:------:|:--------:|:------:|:------:|:-------:|:-------:|
>     | Gowalla  | 0.1483 |  0.1435  | 0.1471 | 0.1474 | 0.1523  | 0.1543  |
>     |  Amazon  | 0.0616 |  0.0631  | 0.0694 | 0.0703 | 0.0833  | 0.0877  |
>     |  Tmall   | 0.0523 |  0.0511  | 0.0567 | 0.0570 | 0.0590  | 0.0658  |
>     | Echonest | 0.0996 |  0.1168  | 0.1340 | 0.1343 | 0.1682  | 0.1842  |
>
>
>     The results demonstrate that the proposed importance-resampling-based sampler, BIR, outperforms all the suggested baselines. The candidate set for sampling of the three baseline methods is the entire corpus, which is much larger than that in BIR. BIR still shows competitive performances with the limited sample set. Another difference lies in that the baseline methods, DNS, LambdaFM and PRIS, optimize the BPR loss whereas BIR is trained using the sampled softmax loss, which may result in a significantly different performance.

---

> > ### Comment · Reviewer_mWFy · 2022-08-08
> > **Discussion on further feedback from authors.**
> >
> > Dear Authors:
> >
> > Great thanks for the detailed response to the raised questions. After reading the reply, I decide not to change my score because of the following concerns:
> >
> > 1. It's a good trial to explain the false negative sampling issue by checking the sampling probability of ground truth recommendations in the test set. However, the conclusion is counterintuitive to many pioneering research. It's not convincing enough because of using the exact probability but the relative value to the other items. It's a natural thing that the more number of items it is, the smaller the average softmax probability will be. Taking the Gowalla as an example, if we take a negative sample from a uniform distribution, the sampling probability is 0.0024%. According to the given statistics by the authors, average sampling probability from a softmax distribution is twenty times over the uniform distribution. Softmax distribution is a skew distribution, but it will become a smoothing distribution as the increasing number of items. Therefore, the statistics shown in the table can not provide convincing evidence to show that the proposed method can deal with the false negative issue. Moreover, if a recommendation method can not give a large probability to the ground truth item, how it's possible to achieve superior ranking performance?
> >
> > 2. Negative samples can be sampled from different distributions [2]. In the second raised concern, I'm actually caring about why sampling from a softmax distribution in this work is superior to the methods which depends on as sampling distribution p(j |u,i) such as LambdaFM, WARP, also those methods from the variants of softmax distributions, like DNS, PRIS, Self-adversarial? While, it's difficult to find the answer from the response.
> >
> > 3. Thanks for making further experiments to check the performance of more baselines. However, the experimental results shown in the table above have conflicts with many pioneering works. It's difficult to believe that an adaptive sampling methods (DNS, LambdaFM) performs worse than methods (SSL, SSL-POP) sampling from static distribution (uniform or popularity-based distribution) in the datasets like Echonest, Amazon.
> >
> > In summary, this work should spend more words on clarifying the motivations and double check the setting of the implementation of the baselines to make the experimental results convincing enough to support the claims in this work.
> >
> > References:
> > [1] One-vs-Each Approximation to Softmax for Scalable Estimation of Probabilities, NIPS 2016.
> > [2] Negative Sampling for Contrastive Representation Learning: A Review, IJCAI 2022.

---

> > > ### Author Response · Authors · 2022-08-09
> > > **Reply to discussion(2/2)**
> > >
> > > **How to deal with false negative**
> > >
> > > The previous analysis tells us that we are optimizing NDCG in the training set. According to the machine learning generalization theory, NDCG can be guaranteed in the test set. Therefore, from this theoretical perspective, we can answer that our method essentially deals with false negative issues.
> > >
> > > From another perspective, according to our provided statistics, the sampling probability of positive items in the test set is extremely low. The sampling probability becomes smaller with the increasing number of items. Since the negative items are randomly sampled according to the probability, these positive items in the test set are rarely sampled as negatives. In other words, false negative cases are hardly encountered in the training process. Intuitively speaking, the occasional occurrences of these cases make little effect on the generalization performance. This intuition is also consistent with the aforementioned theoretical results.
> > >
> > >
> > > **Why adaptive sampling methods (DNS, LambdaFM) perform worse than methods (SSL, SSL-Pop)** **in some datasets**
> > >
> > > DNS and LambdaFM first sample some items according to the uniform distribution, within which the top items are more likely to be sampled as negatives. In other words, DNS and LambdaFM learn to rank positive items higher than **the top item** within a sampled item pool from the uniform distribution. Different form DNS and lambdaFM, when optimizing the sampled softmax loss (the loss $\ell(u,i)=-s(u,i)+\log \sum_{j\in \mathcal{C} \cup \{ i \}} \exp s(u,j)$) with the uniform sampler, the model should try to rank positive items in front of **all items** in the sampled item pool from the uniform distribution. Therefore, optimizing the sampled softmax loss with the uniform sampler can be considered as adaptive as DNS and LambdaFM. Intuitively speaking, compared to DNS and LambdaFM, optimizing the sampled softmax loss with the uniform sampler can learn the model better, thanks to the following two reasons.
> > > + Since more items are used for optimization, their related parameters are simultaneously updated in each batch.
> > > + According to the previous analysis, it essentially optimizes NDCG in the training set, thus it probably outperforms DNS and LambdaFM in the test set with respect to NDCG and other ranking-oriented metrics.
> > >
> > > Therefore, when the uniform sampler is used, optimizing the sampled softmax loss usually leads to better recommendation performance.
> > >
> > > However, SSL and SSL-Pop in this paper are slightly different. In particular, SSL-Pop is the same as the one optimizing the sampled softmax loss with the popularity-based sampler with respect to bias correction, but they differ in whether sampling items within the batch or not. Though inbatch sampling only considers items in the batch, random shuffle of the training dataset in each epoch makes their difference negligible as long as the model is trained for a sufficiently large number of epochs. *Therefore, combined with the analysis in the previous part, SSL-Pop can perform better than DNS and LambdaFM*.  SSL slightly differs in the removal of bias correction from SSL-Pop, so that SSL can not perform as well as SSL-Pop. However, due to inheriting other advantages of SSL-Pop, SSL can still probably perform better in some cases.
> > >
> > > We have carefully checked the code, and did not discover bugs or issues. The source code of this project is also uploaded as supplementary materials. The reviewers are welcomed to check. Here, we also provide more implementation details about DNS and LambdaFM. In particular, within each batch of size $B$, each positive user-item pair is compared with $B$ negatives, each of which is obtained by the sampler of either DNS or LambdaFM (candidate item size is 5). The algorithms are trained by adam with a fixed learning rate of 0.01 within 100 running epochs where the embedding size is 32. The weight decay is tuned over {$0.01,0.001,0.0001,0.00001$}for DNS and LambdaFM with respect to the performance of NDCG\@10 in the validation dataset.

---

> > > > ### Comment · Reviewer_mWFy · 2022-08-09
> > > > **Response to the author feedback**
> > > >
> > > > Dear Authors:
> > > >
> > > > Thank you so much for further feedback to clarify the concerns about the false negative issue and the details about the implementation of adaptive samplers like DNS, LambdaFM. It's good to see more justification that helps a lot to dig more insight about this work. Inbatch negative sampling raises unique challenges comparing to sampling from the whole corpus. This work presents an alternative way to explore the capacity of ranking optimization by leveraging the samples shared in the batch data. Most of my concerns have been addressed, even we have a difference of opinion on some issues.  It's interesting to see that the performance of the proposed method has close relation to the batch size. I think it deserves more attention on designing efficient inbatch sampling method. After reading the rebuttal and checking out the code, I'd like to turn to accept this work.
> > > >
> > > > Minor Issues:
> > > > 1. Please explain what is the meaning of SSL in Section 4.1.2? It's better to give the complete name when it shows up at the first time.

---

> > > ### Author Response · Authors · 2022-08-09
> > > **Reply to discussion(1/2)**
> > >
> > >
> > > **Why sampling from the softmax distribution and why the softmax sampler better**
> > >
> > > According to the theoretical work[1] from Google research,  softmax cross entropy loss is a bound on mean Normalized Discounted Cumulative Gain (NDCG) in log-scale when working with binary ground-truth labels. These theoretical results suggest that optimizing the softmax cross entropy loss is an indirect attempt at optimizing NDCG when given binary relevance judgments. Therefore, the recommender models trained with implicit feedback can achieve superior recommendation accuracy in terms of NDCG and other similar ranking metrics when optimizing the softmax cross entropy loss. Assuming user $u$ has interacted with item $i$, the softmax cross entropy loss is formulated as follows:
> > > $$
> > > \ell(u,i)= - s_\theta(u,i) + \log\sum_{j=1}^N \exp (s_\theta (u,j))
> > > $$
> > > where $s_\theta(u,i)$ indicates the preference score of user $u$ for item $i$, and $N$ denotes the number of items. It is easily observed that the computational cost of the loss scales linearly with $N$. To motivate how to reduce the time cost, we first derive the gradient of the loss w.r.t to the parameter $\theta$ as follows:
> > > $$
> > > \nabla_\theta \ell(u,i)=-\nabla_\theta s_\theta(u,i)+\sum_{j=1}^N P(j|u) \nabla_\theta s_\theta(u,j)
> > > $$
> > > where $P(j|u)=\frac{\exp(s_\theta(u,j))}{\sum_k \exp(s_\theta(u,k))}$ denotes the softmax distribution. *From this equation, we can see that in order to reduce the time cost of parameter update, we can sample some items from the softmax distribution $P(j|u)$ and then estimate the gradient with these samples. We can easily observe that the estimated gradient is unbiased and of a low variance[2].  The unbiasedness and low-variance gradient could make the optimization converge to optimum as fast as possible. Other samplers lead to either biased gradients or unstable gradients, so the converged solutions should be of lower quality than the solutions with the softmax sampler. The characteristics of the softmax sampler motivate why to sample from the softmax distribution.* Note that based on the gradient estimation, we can derive the sampled softmax loss[3,7]. The optimization of the sampled softmax loss for recommender system is now widely-studied by Ed. Chi's team[4,5], as well as Steffen Rendle[6] from Google, reporting the superiority over other baselines.
> > >
> > >
> > >
> > > **Recommendation method should give a large probability to the ground truth item**
> > >
> > > Yes. It should. But we have to point out the differences between sampling and retrieval.  Sampling items from the softmax distribution is used for training while retrieving items with large probabilities is used for top-k recommendation. Please note that sampling is stochastic while retrieval is deterministic. Retrieval only cares about the relative of probability to the other items while sampling concerns the absolute probability of items themselves.  This is not contradictory.
> > >
> > >
> > >
> > > References
> > >
> > > [1] Bruch, S., Wang, X., Bendersky, M., & Najork, M. (2019, September). An analysis of the softmax cross entropy loss for learning-to-rank with binary relevance. In *Proceedings of the 2019 ACM SIGIR international conference on theory of information retrieval* (pp. 75-78).
> > >
> > > [2] Owen, A. B. (2013). Monte Carlo theory, methods and examples.
> > >
> > > [3] Bengio, Y., & Senécal, J. S. (2003, January). Quick training of probabilistic neural nets by importance sampling. In *International Workshop on Artificial Intelligence and Statistics* (pp. 17-24). PMLR.
> > >
> > > [4] Yi, X., Yang, J., Hong, L., Cheng, D. Z., Heldt, L., Kumthekar, A., ... & Chi, E. (2019, September). Sampling-bias-corrected neural modeling for large corpus item recommendations. In *Proceedings of the 13th ACM Conference on Recommender Systems* (pp. 269-277).
> > >
> > > [5] Yang, J., Yi, X., Zhiyuan Cheng, D., Hong, L., Li, Y., Xiaoming Wang, S., ... & Chi, E. H. (2020, April). Mixed negative sampling for learning two-tower neural networks in recommendations. In *Companion Proceedings of the Web Conference 2020* (pp. 441-447).
> > >
> > > [6] Blanc, G., & Rendle, S. (2018, July). Adaptive sampled softmax with kernel based sampling. In *International Conference on Machine Learning* (pp. 590-599). PMLR.
> > >
> > > [7] Bengio, Y., & Senécal, J. S. (2008). Adaptive importance sampling to accelerate training of a neural probabilistic language model. *IEEE Transactions on Neural Networks*, *19*(4), 713-722.

---

> ### Comment · Area_Chair_dw6u · 2022-08-07
> **concerns**
>
> Are you convinced by the clarifications, in particular about the comparisons you asked for?

---

### Official Review · Reviewer_mU7B · 2022-07-12

**Rating:** 8
**Confidence:** 5
**Soundness:** 3 good
**Presentation:** 4 excellent
**Contribution:** 4 excellent

**Summary:**

This paper studies an important research problem of in-batch negative sampling for recommender systems, which aims to efficiently train recommender retrievers (also known as recall-stage models) by choosing the items in the mini-batch as negatives to estimate the softmax function. This is also highly practical and thus widely adopted in today’s industrial recommender systems. The authors first analyze and discuss two main weaknesses of existing solutions. One is a huge bias of the approximated gradient from the full softmax with the in-batch sampling. The other is a query-independent sample selection strategy that is less effective for convergence of the recommendation quality. To tackle the above two problems, the authors propose to simultaneously eliminate approximation bias caused by in-batch sampling and improve sampling quality by favoring hard negative samples. Specifically, the authors propose to resample the batch items according to the modiﬁed softmax weight over the mini-batch, which offers query-dependent negatives and theoretically provides a more accurate estimation of the full softmax. Moreover, the sampling process is augmented with hard negative samples by caching frequently sampled items. The designed Cache-Augmented In-batch Importance Resampling (shorted as χIR) significantly outperforms state-of-the-art baselines by 3.81%~17.12% on ﬁve public real-world datasets.

**Questions:**

1. The authors have analyzed time complexity in Sec. 3.3 and plotted empirical results in Figure 3(c). Is it possible to provide space complexity results, too?
2. Since updating the cache and occurrences takes O(N) time complexity, and item numbers can be enormous in industrial recommender systems, I suggest considering updating the cache in a less-frequent manner and also analyzing its possible impact on recommendation quality.

**Limitations:**

Yes.

**Strengths And Weaknesses:**

Strengths:
1.    The studied problem is important and interesting. In-batch negative sampling is one of the most prevalent techniques for recall-stage recommendation models in real-world applications. Besides, it is more challenging than the traditional negative sampling problems.
2.    The proposed method is novel. The cooperation of importance resampling and hard sample caching can address the two remaining issues of existing in-batch sampling solutions. It is the first work to achieve query-dependent and hard in-batch negative sampling simultaneously.
3.    The two main designs are technically sound. First, a non-asymptotic bound of the proposed importance in-batch resampling method is derived and analyzed rigorously. Second, the designed cache constantly updated by frequently sampled items can capture more informative negatives.
4.    The authors conduct extensive experiments. State-of-the-art methods are compared on five real-world datasets, and the results show a significant improvement achieved by the proposed method. The ablation studies, applicability studies, and hyper-parameter studies are solid.
5.    Overall, the presentation is nice. I enjoy the reading, and I believe most readers of the NeurIPS community will, too.

Weaknesses:
1.    Table 2 presents many results of the performance comparison. But, there is too much information in a single table. I suggest bolding and underlining the best and second best, respectively.
2.    I appreciate the derivation and analysis of the non-asymptotic bound (Theorem 3.2). However, It would be more informative and instructive if the authors could provide a more thorough analysis of the impacts of batch size and popularity distribution.
3.    A few typos. e.g., “Assum that the gradients of the logits” in Line 162. And y-label for Figure 3(c).

---

> ### Author Response · Authors · 2022-08-02
> **Reply to Reviewer mU7B**
>
> Thank you for your valuable reviews. We will correct the typos in this paper and highlight the results in Table 1. What follows is our response to your comments.
>
> 1. [More Analysis w.r.t. Batch Size and Popularity Distribution]
>     > **Comment**: However, It would be more informative and instructive if the authors could provide a more thorough analysis of the impacts of batch size and popularity distribution.
>
>     According to the theoretical results, with a larger batch size, the upper bound gets smaller, indicating that the estimated gradient is less biased. In terms of the popularity distribution, if the value of the popularity differs greatly, i.e., $\max pop(\cdot) / \min pop(\cdot)$ has a greater value, it would get a larger bias of the gradient. Thank you for your suggestion and these analyses will be added.
> 2. [Space Complexity]
>     > **Comment**: Is it possible to provide space complexity results, too?
>
>     For BIR, it only takes an additional $O(\vert B \vert )$ space for each user to store the index of the sampled item. Thus, for each mini-batch $B$, it takes $O(\vert B \vert \times \vert B \vert)$ space complexity.
>
>     For XIR, it further takes $O(\vert C \vert)$ to save the cached items and $O(N)$ to count the occurrence of the items, where $C$ denotes the cache and $N$ denotes the number of items. Thus, the space complexity is $O(\vert B \vert \times \vert B \vert + \vert C \vert + N)$.
> 3. [Cache Update]
>     > **Comment**: I suggest considering updating the cache in a less-frequent manner and also analyzing its possible impact on recommendation quality.
>
>     Thank you for your advice to update the cache less frequently to reduce the time cost.

---

### Meta-Review · Area_Chair_dw6u · 2022-08-22

**Recommendation:** Accept
**Confidence:** Certain

**Metareview:**

The idea of more representative mini-batches sounds like a natural extension of the work done on stratified sampling. The reviewers were convinced the idea is both new and effective on real data. In particular, the discussion with mWFy clarified that this work is alternative way to explore the capacity of ranking optimization by leveraging the samples shared in the batch data by comparing it to other relevant work.

**Award:**

No

---

### Decision · Program_Chairs · 2022-09-14

Accept